# Engram-specific transcriptome profiling of contextual memory consolidation

Priyanka Rao-Ruiz [1,2], Jonathan J. Couey[1], Ivo M. Marcelo[1,3], Christian G. Bouwkamp[1], Denise E. Slump[1], Mariana R. Matos[2], Rolinka J. van der Loo[2], Gabriela J. Martins[3,4], Mirjam van den Hout [5], Wilfred F. van IJcken [5], Rui M. Costa[3,4], Michel C. van den Oever [2] & Steven A. Kushner [1]

Sparse populations of neurons in the dentate gyrus (DG) of the hippocampus are causally implicated in the encoding of contextual fear memories. However, engram-specific molecular mechanisms underlying memory consolidation remain largely unknown. Here we perform unbiased RNA sequencing of DG engram neurons 24 h after contextual fear conditioning to identify transcriptome changes specific to memory consolidation. DG engram neurons exhibit a highly distinct pattern of gene expression, in which CREB-dependent transcription features prominently ($P = 6.2 \times 10^{-13}$), including *Atf3* ($P = 2.4 \times 10^{-41}$), *Penk* ($P = 1.3 \times 10^{-15}$), and *Kcnq3* ($P = 3.1 \times 10^{-12}$). Moreover, we validate the functional relevance of the RNAseq findings by establishing the causal requirement of intact CREB function specifically within the DG engram during memory consolidation, and identify a novel group of CREB target genes involved in the encoding of long-term memory.

[1] Department of Psychiatry, Erasmus MC University Medical Center, Rotterdam 3015 GD, The Netherlands. [2] Department of Molecular and Cellular Neurobiology, Center for Neurogenomics and Cognitive Research, Amsterdam Neuroscience, Vrije Universiteit Amsterdam, Amsterdam 1081 HV, The Netherlands. [3] Champalimaud Neuroscience Programme, Champalimaud Centre for the Unknown, Lisbon 1400-038, Portugal. [4] Department of Neuroscience, Zuckerman Mind Brain Behavior Institute, Columbia University, New York 10027 NY, USA. [5] Center for Biomics, Erasmus MC University Medical Center, Rotterdam 3015 GD, The Netherlands. Correspondence and requests for materials should be addressed to M.Oever. (email: michel.vanden. oever@vu.nl) or to S.A.K. (email: s.kushner@erasmusmc.nl)

Fear memories are encoded and stored in the brain by sparse ensembles of neurons collectively termed as memory engrams or traces. Selective ablation[1] or optogenetic silencing[2] of engram neurons results in a deficit of conditioned fear responding, while targeted activation of molecularly tagged engrams is sufficient to elicit memory expression[3]. In particular, the dentate gyrus (DG) of the hippocampus is critical to the encoding of the contextual representation associated with fear memories, wherein an estimated 2–4% of DG neurons exhibit modulated activity during retrieval of contextual fear memories[4].

The cellular mechanisms of memory allocation to engram cells has been carefully investigated, revealing the intrinsic excitability of dentate neurons as a critical determinant underlying their recruitment into a memory engram[5,6]. Once allocated, the successful consolidation of memory requires a dynamic time-dependent process of gene transcription[7] and protein translation[8]. Recent technological advancements have made it possible to examine early transcriptional changes in sparsely distributed ensembles due to the rapid expression of immediate early genes (IEGs) after an activity-inducing experience[9]. However, the enduring molecular dynamics necessary for memory consolidation within engram cells encoding contextual fear memories have yet to be revealed due the transient nature of most IEGs.

Here, demonstrate that the IEG, *Activity Regulated Cytoskeleton Associated Protein* (*Arc*), is selectively and persistently expressed in DG engram cells after fear conditioning. This sustained expression of *Arc* enabled us to examine the differential transcriptional profile of DG memory-trace neurons compared to their nonactivated neighbors, 24 h after fear conditioning. Our findings revealed genome-wide alterations in the neuronal transcriptome of engram cells during contextual fear memory consolidation. In particular, unbiased upstream analysis revealed the CREB network to be activated exclusively in engram neurons after fear conditioning (FC), a finding causally validated by manipulating CREB function specifically in engram neurons.

## Results

**Sustained activation of *Arc* after fear conditioning.** In order to visually label neurons activated during the encoding of a fear memory, we made use of the *Arc*::dVenus mouse line[10]. In this system, the expression of a destabilized fluorescent reporter (dVenus) is coupled to the promoter of the IEG *Activity Regulated Cytoskeleton Associated Protein* (*Arc*)[10] (Supplementary Fig. 1a), a well-established marker of recent neuronal activity[11]. FC leads to the formation of a robust contextual fear memory (Supplementary Fig. 1b, c) with concordant dVenus expression in a sparse population of neurons distributed along the rostrocaudal axis of the DG (Supplementary Fig. 1d), consistent with prior observations of *Arc* expression in the DG following novel experience[12]. We observed high co-localization between endogenous Arc protein, the *Arc*::dVenus reporter, and the proto-oncogene c-Fos 90 min after FC (P[Fos$^+$|Arc$^+$] = 85.2 ± 1.3%, P[Arc$^+$|Fos$^+$] = 96.3 ± 0.7%, P[Fos$^+$|dVenus$^+$] = 82.1 ± 2.6%, P[dVenus$^+$|Fos$^+$] = 82.7 ± 4.1%) (Supplementary Fig. 2), confirming that Arc and Fos tag a highly overlapping population of DG engram neurons.

We next aimed to characterize the temporal activation profile of DG memory engram neurons by quantifying *Arc*::dVenus expression at successive time-points after FC (Fig. 1a). The number of dVenus$^+$ cells exhibited a rapid (within 1 h) and sustained (up to 24 h) increase following training (baseline: 10.54 ± 1.96 cells per 1.3 mm$^2$, 1 h: 30.13 ± 0.69 cells per 1.3 mm$^2$, 5 h: 34.96 ± 1.66 cells per 1.3 mm$^2$, 8 h: 29.26 ± 1.48 cells per 1.3 mm$^2$, 14 h: 31.99 ± 1.91 cells per 1.3 mm$^2$, 24 h: 36.98 ± 4.14 cells per 1.3 mm$^2$) (Fig. 1b, c). This sustained hippocampal *Arc*::dVenus activation

was specific to the DG and not observed in the CA1 or CA3 subregions, in which dVenus$^+$ cells were robustly observed at 5 h, but no longer at 24 h after training (Supplementary Fig. 3).

Next, we explored whether the temporal stability over 24 h in the number of DG dVenus$^+$ cells resulted from the recruitment of a stable ensemble with sustained dVenus$^+$ expression, or whether the population of dVenus$^+$ cells—although maintained as a constant overall number—is dynamically changing. In order to distinguish between these possibilities, we performed in vivo microendoscopic imaging to monitor dVenus expression in DG cells over the 24 h time course (Fig. 1d). Consistent with a largely stable population, we found that dVenus$^+$ cells exhibited persistent expression over time (Fig. 1e–g). In particular, 79.8% of dVenus$^+$ neurons at 5 h were also dVenus$^+$ at 24 h (Fig. 1e). Conversely, 73.5% of dVenus$^+$ cells at 24 h were also dVenus$^+$ at 5 h (Fig. 1f). Finally, we confirmed that the sustained expression of *Arc*::dVenus at 24 h was due to enduring expression of endogenous Arc by performing a double immunostaining. As expected, we observed a higher level of co-localization between Arc and dVenus in the DG of FC animals compared to home-cage (HC) or no-shock (NS) controls (P[Arc$^+$|dVenus$^+$]; HC: 36.5 ± 12.4%, NS: 58.8 ± 2.1%, FC: 84.10 ± 1.3%) (Fig. 1h, i). Lastly, we performed a longitudinal series of quantifications of the co-localization between endogenous Arc and *Arc*::dVenus reporter after conditioning (P[Arc$^+$|dVenus$^+$]; 1 h: 81.3 ± 1.7%, 5 h: 71.6 ± 0.5%, 14 h: 83.7 ± 0.9%) (Supplementary Fig. 4).

Taken together, these data confirm that *Arc* exhibits sustained expression for at least 24 h in DG fear memory neuronal ensembles.

**The engram has a distinct transcriptome during consolidation.** Memory consolidation is a dynamic process requiring several waves of gene transcription, with a delayed wave being necessary for the persistence of long-term memory[13]. However, investigations of the molecular underpinnings of memory consolidation in engram cells have thus far been limited by: (1) the transient nature of neuronal IEG expression, and (2) the sparse distribution of the engram. Therefore, the sustained expression of *Arc* within the DG engram presented us with the unique opportunity to query enduring molecular changes. Using fluorescence-guided cell aspiration, we performed RNA sequencing from neighboring dVenus$^+$ and dVenus$^-$ cells to examine their differential gene expression profiles 24 h after FC. From each animal, the contents of 10 dVenus$^+$ and 10 neighboring dVenus$^-$ DG cells were aspirated using a modified approach for pulling nucleated patches[14,15] (Fig. 2a). Full length cDNA was generated from each ten-cell sample using the SmartSeq 2[16] protocol. Illumina HiSeq Rapid v2 sequencing chemistry was utilized to generate a minimum of 10 M aligning reads per sample. A total of 16 paired samples from FC, 4 paired samples from NS and 4 paired samples from HC were collected, of which 4 FC paired samples and 1 HC paired sample did not pass quality control and were excluded from further analyses (Supplementary Data 1). In total, 11,802 genes passed quality control and were subjected to multi-dimensional scaling and clustering. Regularized log counts of a panel of known DG granule cell-enriched genes[17] further confirmed the cell type-specificity (Supplementary Fig. 5). Sample-to-sample principal component analysis for the top 100 genes across all conditions revealed that PC1 scores (18% variance) distinguished samples based on cell activation (dVenus$^+$ vs. dVenus$^-$ neurons) (Fig. 2b, Supplementary Data 2). Moreover, PC2 scores (11% variance) separated samples based on training history, with dVenus$^+$ cells from the FC group of 12 independent replicates splitting away from dVenus$^+$ cells of the NS and HC groups (Fig. 2b). PCA analysis of the top 500 genes also resulted

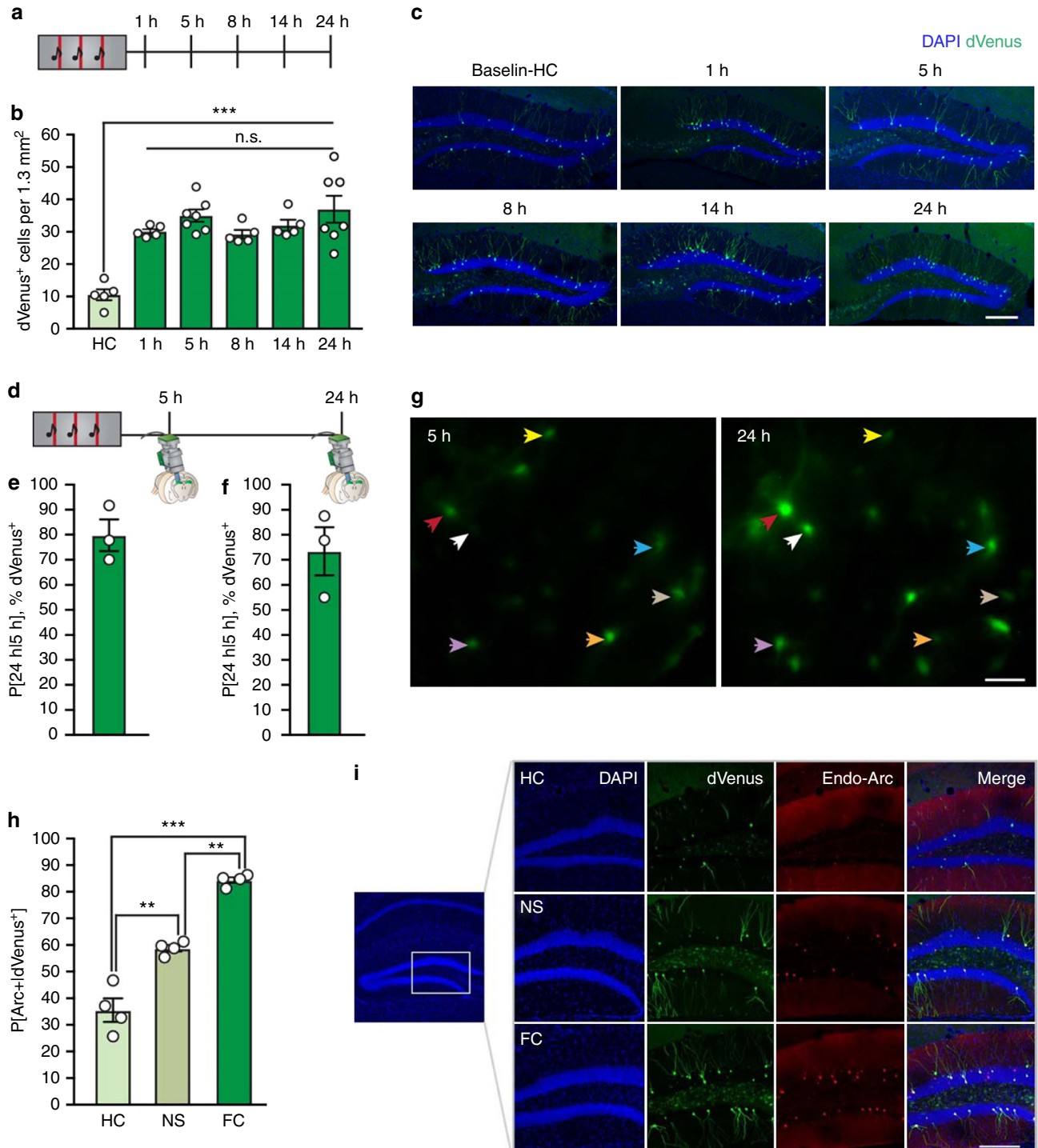

**Fig. 1** Activity-dependent, sustained expression of *Arc*::dVenus in DG granule cells. **a** Experimental setup. *Arc*::dVenus mice were fear conditioned and the number of dVenus+ cells was measured in the DG at successive time-points; 1 h ($n = 5$), 5 h ($n = 7$), 8 h ($n = 5$), 14 h ($n = 5$) and 24 h ($n = 7$), after training. Home-cage (HC) controls ($n = 5$) serve as a baseline. **b** Number of dVenus+ cells per 1.3 mm² section in the DG, at specific time-points after fear conditioning. Analysis of variance: effect of training history over baseline (HC): $F_{(1,33)} = 13.102$, $P = P = 1.0 \times 10^{-5}$; post hoc LSD: HC vs. 1 h: $P = 2.2 \times 10^{-5}$, HC vs. 5 h: $P = 1.9 \times 10^{-5}$, HC vs. 8 h: $P = 4.0 \times 10^{-5}$, HC vs. 14 h: $P = 6.0 \times 10^{-6}$, HC vs. 24 h: $P = 4.5 \times 10^{-8}$. **c** Representative images of the DG from fear conditioned mice at each successive time-point after fear conditioning. Scale bar: 200 μm. **d** Animals were implanted with microendoscopes to longitudinally monitor in vivo dVenus fluorescence in the DG ($n = 3$). **e** Percentage of dVenus+ cells at 5 h that also express dVenus 24 h after fear conditioning. **f** Percentage of dVenus+ cells at 24 h that also expressed dVenus 5 h after fear conditioning. **g** Representative microendoscopy images of dVenus+ cells at 5 and 24 h. Colored arrows indicate cells expressing dVenus at both time-points. Scale bar: 100 μm. **h** Percentage of dVenus+ cells in the DG that also express endogenous Arc in home-cage controls (HC, $n = 4$), no shock controls (NS, $n = 4$) or fear conditioned animals (FC, $n = 4$). Multivariate analysis of variance: $F_{(2,12)} = 40.2$, $P = 0.0003$, post hoc LSD: HC vs. NS: $P = 0.006$, HC vs. FC: $P = 0.0001$, NS vs. FC: $P = 0.003$. **i** Representative images demonstrating co-expression of endogenous Arc and dVenus. *$P < 0.05$, **$P < 0.01$, ***$P < 0.001$. Data are presented as mean ± SEM. Scale bar: 200 μm. Source data are provided as a Source Data file

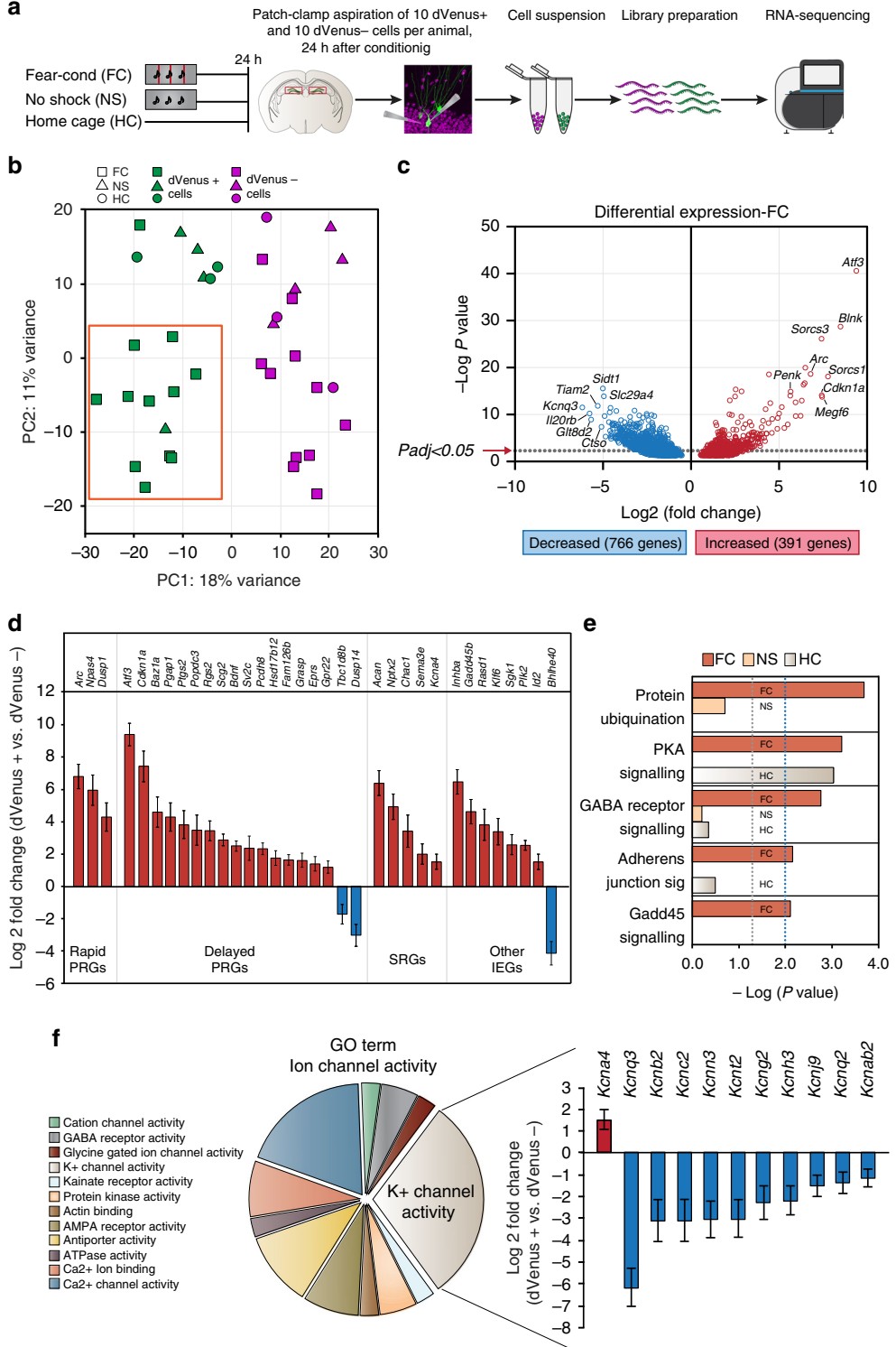

in a similar distinction of cells based on their activation and training history, indicative of a transcriptome robustly unique to fear memory engram cells (Supplementary Fig. 6).

Differential gene expression analysis (DeSeq2[18]) using a group-wise paired-sample design (dVenus+ vs. dVenus−) revealed transcriptome changes specific to dVenus+ cells (Supplementary Data 3) in all three experimental groups (Supplementary Fig. 8b–d). A total of 1157 genes in the FC group (Fig. 2c), 175 in the NS group (Supplementary Fig. 7a), and 638 genes in the HC group (Supplementary Fig. 7b) exhibited differential regulation between dVenus+ and dVenus− neurons (false-discovery rate (FDR) corrected $P$ value < 0.05 with absolute $\log_2$ fold change > 1.0). Of these, 10 genes were differentially expressed in both the HC and NS groups, 92 genes in both the HC and FC groups, 26 genes in both the NS and FC groups, and 2 genes in all three experimental groups (Supplementary Fig. 7c–e). Variability between libraries was addressed using a sample-to-sample correlation matrix (Supplementary Fig. 8a). Notably, the majority

**Fig. 2** Fear conditioning induces a unique transcriptional profile in DG engram cells. **a** Experimental setup. Nucleated patch aspiration was performed 24 h after fear conditioning (FC, $n = 12$ biological replicates), context-only exposure (NS, $n = 4$ biological replicates), or naïve home-cage controls (HC, $n = 3$ biological replicates). **b** Sample-to-sample principal component analysis. PC1 scores separated samples by state of activation (dVenus+ [green] vs. dVenus− [magenta]) across all experimental groups, while PC2 separated samples based on their training history (fear conditioned group [FC] vs. naïve home-cage [HC] and no-shock [NS] controls). Orange rectangle delineates the corresponding PC1/PC2 isolated quadrant. **c** Differential expression between dVenus+ and dVenus− cells for all genes with a raw $P < 0.05$. Dotted line indicates $P_{adj} < 0.05$ (FDR corrected). Genes that are upregulated in dVenus+ cells are in red, and genes that are downregulated in dVenus+ cells are in blue . The top 7 up and downregulated genes along with the total number of regulated genes with $P_{adj} < 0.05$ are labeled. **d** Log$_2$ fold change of a panel of known activity regulated genes between dVenus+ and dVenus− cells 24 h after fear conditioning. PRGs primary response genes, SRGs secondary response genes. Data are presented as mean ± SEM. **e** Functional pathway enrichment with $P < 0.01$ of differentially expressed genes in the FC group. The enrichment of these pathways in the NS and HC groups is plotted alongside the FC group. Gray dotted line indicates significance threshold set at $-\log_{10} P > 1.3$ ($P < 0.05$, Fisher's exact test), and blue dotted line indicates significance threshold set at $-\log_{10} P > 2$ ($P < 0.01$, Fisher's exact test). **f** Gene ontology (GO) analysis of molecular function revealed Ion channel activity as overrepresented in the FC group (GO:0005216, $P = 2.7 \times 10^{-5}$, FDR corrected Fisher's exact test). Of the 40 genes in this GO class, 11 were K+ channels. The genes of these K+ channels are plotted in the right panel as a log$_2$ fold change between dVenus+ and dVenus− cells. Data are presented as mean ± SEM

of genes identified 24 h after FC were not identified in transcriptomic analyses of (1) whole hippocampus 1 or 24 h after seizure induction[19], (2) activated DG granule cells 1 h after novelty exposure[9], (3) whole hippocampus 5 min, 30 min, 1 h or 4 after FC[20], or (4) activated ensembles from the temporal association cortex 6 h after auditory FC[21] (Supplementary Data 4).

As expected, endogenous *Arc* was highly upregulated in dVenus+ cells compared to dVenus− cells across all experimental groups (FC: Log$_2$ fold change = 6.79, $P = 2.3 \times 10^{-19}$, $P_{adj} = 4.7 \times 10^{-16}$, NS: Log$_2$ fold change = 8.40, $P = 9.6 \times 10^{-11}$, $P_{adj} = 4.5 \times 10^{-7}$, HC: Log$_2$ fold change = 8.12, $P = 2.3 \times 10^{-13}$, $P_{adj} = 5.1 \times 10^{-10}$) (Fig. 2c, Supplementary Fig. 7a, b, Supplementary Data 3). We next asked whether the sustained activation profile that we observed for *Arc* was unique to this IEG, or whether other known activity regulated genes (ARGs) were also persistently expressed in engram cells. Thirty-four ARGs[9,22–25] were differentially expressed (Fig. 2d), of which only *Arc* was also regulated in HC and *Arc*, *Dusp14*, *Nptx2*, *Inhba*, and *SgK1* were also regulated in the NS condition. Eighteen of the 34 activity related genes identified were delayed primary response genes that belong to a second wave of plasticity-related genes that require sustained activity, de novo translation and cell signaling pathway induction[24]. In contrast, other well-described learning-associated IEGs (*Fos*, *Junb*, *Homer1*, *Egr1*, *Erg2*, *Egr3*, *Egr4*) previously shown to exhibit prominent upregulation immediately following salient novel behavioral experience[9], were unaltered in DG engram neurons 24 h after FC (Supplementary Data 3).

The most significantly regulated gene in the FC group was the transcription factor *Atf3* (670-fold upregulated in dVenus+ engram, log$_2$ fold change = 9.38, $P = 2.4 \times 10^{-41}$, $P_{adj} = 2.5 \times 10^{-37}$) (Fig. 2c), previously implicated in experience-dependent actin structural plasticity[26]. Accordingly, we investigated the longitudinal 24 h time-course of postconditioning Atf3 protein expression. Bimodal peaks of Atf3+ cells were observed at 5 and 24 h after FC (baseline: 2.43 ± 1.97 cells per 0.6 mm$^2$, 1 h: 4.91 ± 0.33 cells per 0.6 mm$^2$, 5 h: 10.53 ± 0.82 cells per 0.6 mm$^2$, 14 h: 4.33 ± 1.84 cells per 0.6 mm$^2$, 24 h: 11.52 ± 1.77 cells per 0.6 mm$^2$) (Supplementary Fig. 9a–c), indicative of a dynamic expression profile consistent with transient waves of structural plasticity thought to underlie long-term memory formation[27,28]. Because few Atf3+ cells were positively labeled, our estimate of the proportion of dVenus+ cells expressing Atf3 was less reliable (Supplementary Fig. 9d). The discrepancy between the fold-change of Atf3 RNA compared to the protein abundance measured by immunolabeling is likely a technical limitation of the antibody quality, absolute Atf3 RNA abundance, and/or regulation of Atf3 RNA translation[29]. Consistent with this view, we consistently observed, across all experimental conditions, that nearly every measured Atf3+ cell was dVenus+ (Supplementary Fig. 9e). In

addition, two different vacuolar protein sorting 10 (VPS10) domain-containing receptor family members, *Sorcs1* (Log$_2$ fold change = 7.77, $P = 8.3 \times 10^{-19}$, $P_{adj} = 1.3 \times 10^{-15}$) and *Sorcs3* (Log$_2$ fold change = 7.41, $P = 7.4 \times 10^{-27}$, $P_{adj} = 2.6 \times 10^{-23}$), vacuolar protein sorting 10 (VPS10) domain-containing receptor family members with known functions in AMPA receptor trafficking[30,31], exhibited a 220- and 170-fold upregulation respectively, in dVenus+ engram neurons (Fig. 2c). *Penk*, encoding the endogenous opioid polypeptide hormone proenkephalin was 50-fold upregulated (Log$_2$ fold change = 5.66, $P = 1.3 \times 10^{-15}$, $P_{adj} = 1.0 \times 10^{-12}$). Furthermore *Acan*, encoding the integral extracellular matrix protein aggrecan, was also significantly upregulated by 84-fold, consistent with recent hypotheses about the function of perineuronal nets in the storage of long-term memories (Log$_2$ fold change = 6.39, $P = 4.5 \times 10^{-17}$, $P_{adj} = 5.2 \times 10^{-14}$) (Supplementary Data 3).

Ingenuity pathway analysis revealed 3 significantly enriched pathways ($P < 0.01$) in the NS (Supplementary Fig. 10a, Supplementary Data 5) and HC groups (Supplementary Fig. 10b, Supplementary Data 5), and 5 pathways in the FC group (Fig. 2e, Supplementary Data 5). Furthermore, GO analysis of significantly regulated genes revealed no overrepresented functional classes in the HC group or the NS group. In contrast, 2 functional classes were overrepresented in the FC group, receptor binding (GO: 0005102, $P = 8.7 \times 10^{-4}$) and ion channel activity (GO: 0005216, $P = 2.7 \times 10^{-5}$). Notably, of the 40 genes identified in the GO class of ion channel activity, 11 were potassium channels (Fig. 2f) including the voltage-gated K+ channel *Kcnq3*, which was 72-fold downregulated (Log$_2$ fold change = −6.16, $P = 3.1 \times 10^{-12}$, $P_{adj} = 1.3 \times 10^{-9}$) in dVenus+ engram neurons (Fig. 2c, Supplementary Data 3).

**A CREB-dependent network is recruited in engram neurons.** Network analysis of the top 50 differentially regulated genes revealed a CREB-dependent transcriptional network as the predominant contributor, encompassing 22 of 50 genes (44.0%, overlap $P = 6.2 \times 10^{-13}$) and enriched specifically in the FC group (activation z-score = 3.71, $P = 1.09 \times 10^{-12}$) (Fig. 3a, Supplementary Data 6). Of the 22 genes, 16 were robustly upregulated in dVenus+ cells; while 6 were downregulated (Fig. 3b). Using multiplex fluorescent RNAscope in situ hybridizations[32], we validated our RNA sequencing results for three of the top ranked genes identified as part of the CREB network—the upregulated genes *Arc*, *Atf3*, and *Penk*, and also validated the expression of the most significantly downregulated gene (*Kcnq3*) identified in our screen. Together with the *dVenus* reporter transcript, co-expression was quantified in *Arc*+ cells in comparison to their nonactivated neighbors 24 h after FC (Fig. 3c, d). Consistent with the differential gene expression found by

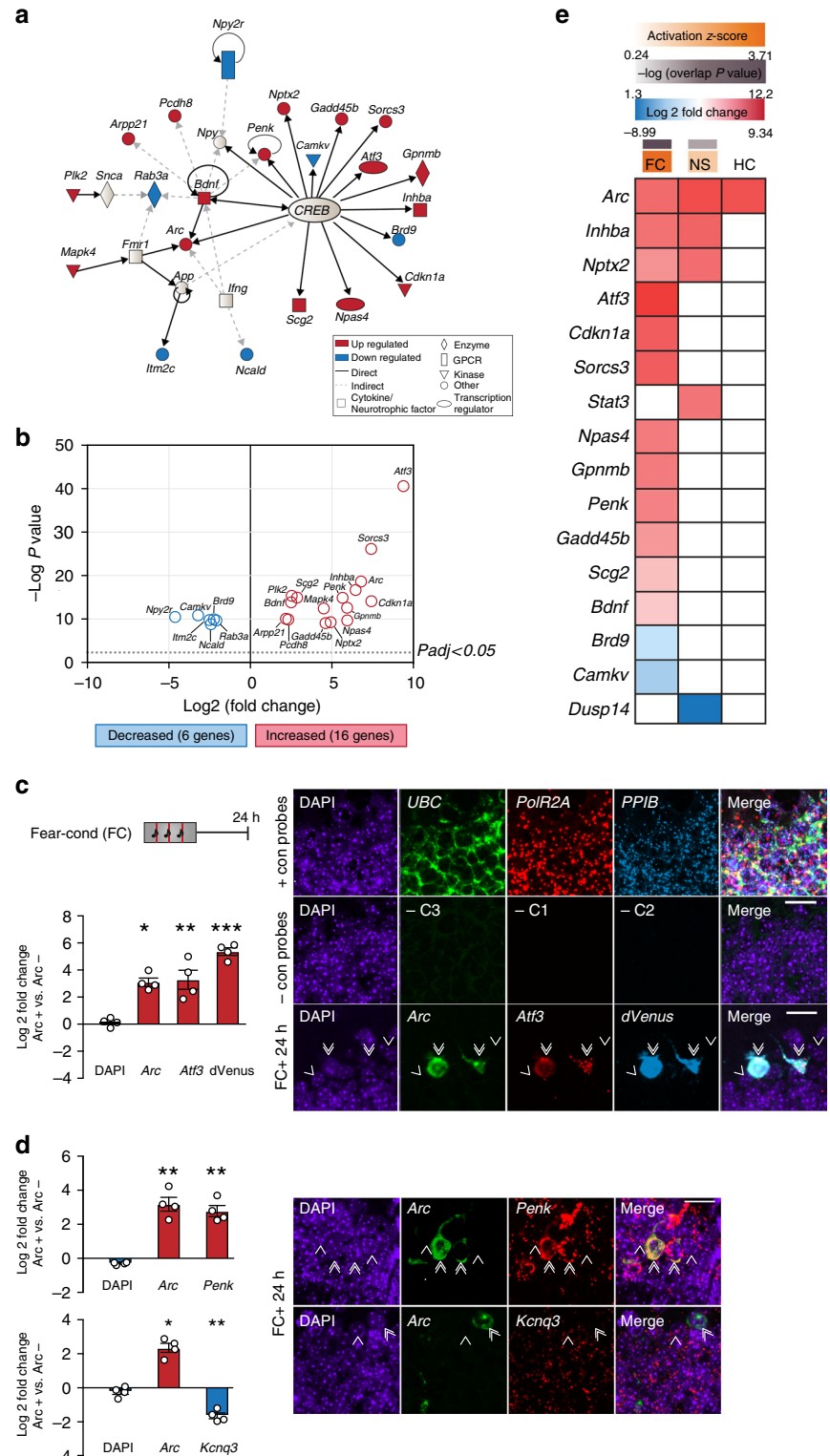

RNA sequencing, *Arc* (Fig. 3c: Log$_2$ fold change = 3.13, $P = 1.9 \times 10^{-2}$, Fig. 3d (upper): Log$_2$ fold change = 3.18, $P = 7.8 \times 10^{-3}$, Fig. 3d (lower): Log$_2$ fold change = 2.37, $P = 3.1 \times 10^{-2}$) (Fig. 3c), *Atf3* (Log$_2$ fold change = 3.02, $P = 7.5 \times 10^{-4}$), *dVenus* (Log$_2$ fold change = 5.62, $P = 6.0 \times 10^{-6}$) (Fig. 3c) and *Penk* (Fig. 3d, upper) (Log$_2$ fold change = 2.96, $P = 2.5 \times 10^{-3}$) were upregulated, while *Kcnq3* (Fig. 3d, lower) (Log$_2$ fold change = −1.5, $P = 6.9 \times 10^{-4}$)

was downregulated in engram cells. In contrast, unbiased upstream analysis showed that the CREB network was not significantly activated in the NS group (activation *z*-score of 1.34, $P = 0.18$) despite a small but significant CREB transcriptional network enrichment (10.0%, overlap $P = 3.5 \times 10^{-3}$) (Fig. 3e). Moreover, with the exception of *Arc*, no other genes regulated by CREB were significantly altered in the HC group. Notably, in

**Fig. 3** Distinct activation of a CREB-dependent network exclusively in DG engram cells. **a** Fear conditioning-induced CREB-dependent gene network activation. Twenty-two of the top 50 significantly regulated genes after FC are part of the CREB network, of which 14 have direct transcriptional regulation. **b** Differential expression between dVenus$^+$ and dVenus$^-$ cells of the 22 genes identified in the CREB network. Dotted line indicates $P_{adj} < 0.05$ (FDR corrected). Red: Genes upregulated, Blue: Genes downregulated, in dVenus$^+$ cells. **c** Multiplex RNA-scope validates the differential expression pattern of *Arc, Atf3*, and dVenus 24 h after fear conditioning. Left: Log$_2$ fold change of fluorescence intensity between *Arc$^+$* and neighboring *Arc$^-$* cells is reported for each gene (*Arc + Atf3 +* dVenus: $n = 4$). Analysis of variance: *Arc*: $F_{(1,7)} = 10.19$, $P = 1.9 \times 10^{-2}$, *Atf3*: $F_{(1,7)} = 39.58$, $P = 7.5 \times 10^{-4}$, dVenus: $F_{(1,7)} = 225.17$, $P = 6 \times 10^{-6}$. Right: Representative images demonstrating positive and negative-control probes as well as co-expression patterns of *Arc* (green), *Atf3* (red), and dVenus (cyan) in the DG of animals. DAPI (blue) labels all cells. 6×. **d** Multiplex RNA-scope experiments to validate the differential expression pattern of *Arc, Penk*, and *Kcnq3* 24 h after fear conditioning. Left: Log$_2$ fold change of fluorescence intensity between *Arc$^+$* and neighboring *Arc$^-$* a cell is reported for each gene (*Arc + Penk*: $n = 4$, *Arc + Kcnq3*: $n = 4$). Analysis of variance: Upper: *Arc*: $F_{(1,7)} = 15.30$, $P = 7.8 \times 10^{-3}$, *Penk*: $F_{(1,7)} = 24.91$, $P = 2.5 \times 10^{-3}$, Lower: *Arc*: $F_{(1,7)} = 7.87$, $P = 3.1 \times 10^{-2}$, *Kcnq3*: $F_{(1,7)} = 40.86$, $P = 6.9 \times 10^{-4}$. Right: Representative images demonstrating co-expression patterns of *Arc* (green) and *Penk* (red), or *Arc* (green), and *Kcnq3* (red) in the DG of animals. DAPI (blue) labels all cells. **c, d** Double arrows: *Arc$^+$* cells, single arrows: neighboring *Arc$^-$* cells. *$P < 0.05$, **$P < 0.01$, ***$P < 0.001$. Data are presented as mean ± SEM. Scale bar: 20 μm. Source data are provided as a Source Data file. **e** Group-wise analysis of significantly regulated genes under direct transcriptional regulation of CREB. The overlap $P$ value measures the enrichment of regulated genes from our data sets, compared to previously identified CREB targets. The activation $z$-score predicts the activation state of the upstream regulator (CREB in this case) based on the log$_2$-fold change values of CREB targets. $z$-scores greater than 2 or smaller than $-2$ are considered significant

contrast to its downstream transcriptional targets, the expression of CREB itself remained unchanged across all conditions (Supplementary Fig. 11). Together, these findings suggest that CREB-dependent transcription functions critically within the DG and specifically within the sparse population of memory engram cells during consolidation.

**Consolidation requires engram-specific CREB transcription.** Finally, we wanted to validate our RNA-sequencing findings and evaluate whether the observed CREB network functions causally within the DG engram during consolidation of contextual fear memory. In order to disrupt CREB-mediated transcription exclusively in engram cells, we utilized *Fos*::tTA transgenic mice[3] in combination with adeno-associated virus (AAV)-mediated gene transfer to selectively express the well-validated dominant-negative CREB$^{S133A}$ transcriptional repressor[33,34] (AAV5-TRE::EGFP-mCREB) in post-training DG neurons activated during FC. This approach couples the *Fos* promoter to the tetracycline-controlled transactivator (tTA), thereby enabling inducible expression of EGFP-mCREB restricted specifically to engram cells (Fig. 4a). In the presence of doxycycline (Dox), tTA mediated transcription of EGFP-mCREB is prevented, whereas in the absence of Dox, FC selectively induces EGFP-mCREB expression in the sparse *Fos$^+$* population of DG engram neurons (Fig. 4b, c). We observed very low expression of EGFP-mCREB in animals that were maintained on Dox and fear conditioned (on-Dox FC) or taken off Dox but not trained (Off-Dox HC). In contrast, mice removed from Dox and fear conditioned (Off-Dox FC) had robust activation in the DG granule cell layer. Moreover, WT mice injected with the TRE::mCREB virus and fear conditioned had negligible expression of EGFP-mCREB in the DG compared to *Fos*::tTA transgenic mice (Supplementary Fig. 12), further demonstrating the tight regulation of mCREB expression.

To validate the efficacy of mCREB in repressing the transcription of identified network genes in DG engram cells, mice injected with either control or EGFP-mCREB vectors were fear conditioned off Dox and the co-expression of *Arc, Atf3*, and *Penk* was evaluated 24 h later in *Arc$^+$* (for control vector) or EGFP$^+$ DG cells, and compared to their nonactivated neighboring cells (Fig. 4d, e). Consistent with the RNA-seq data, *Atf3* (Log$_2$ fold change = 2.68, $P = 3.3 \times 10^{-3}$) and *Penk* (Log$_2$ fold change = 3.04, $P = 6.2 \times 10^{-4}$) were robustly upregulated in *Arc+* cells of mice receiving the control vector, along with *Arc* itself (*Arc* in *Arc + Atf3*: Log$_2$ fold change = 2.29, $P = 3.7 \times 10^{-4}$, *Arc* in *Arc + Penk*: Log$_2$ fold change = 3.67, $P = 8.9 \times 10^{-3}$) (Fig. 4d, e, panels 1 and 3). In contrast, expression of *Arc, Atf3*, and *Penk* was strongly

repressed in EGFP-mCREB$^+$ neurons (EGFP-mCREB in *Arc + Atf3*: Log$_2$ fold change = 2.98, $P = 1.1 \times 10^{-4}$, EGFP-mCREB in *Arc + Penk*: Log$_2$ fold change = 3.08, $P = 6.5 \times 10^{-5}$) (Fig. 4d, e, panels 2 and 4), thereby demonstrating their CREB-dependent transcription 24 h after FC. In addition, at the protein level, the increase in the number of Atf3$^+$ DG neurons observed 24 h after FC was abolished in mice injected with EGFP-mCREB (Supplementary Fig. 9), providing further validation of engram-specific EGFP-mCREB as a robust tool for spatiotemporally-restricted disruption of CREB transcription in vivo.

In order to test whether CREB function is required in the DG engram for consolidation of contextual fear memory, mice were removed from Dox and fear conditioned 48 h later. Immediately after training, mice were returned to Dox to prevent subsequent expression of EGFP-mCREB (Fig. 5a). All animals exhibited a similar increase in freezing after the last US delivery (Fig. 5b). However, mice injected with the mCREB virus exhibited a profound long-term contextual memory deficit when tested 72 h later (Fig. 5c). To examine if mCREB expression in a similar but random population of DG neurons affects consolidation, mice injected with mCREB were taken off Dox during exposure to a novel context, put back on Dox immediately after, and fear conditioned 24 h later. No deficits in memory were observed (Fig. 5d, e) even though the same number of DG cells expressed EGFP-mCREB after either FC or novel context exposure (FC: 37.36 ± 2.77 cells per 0.6 mm$^2$, NC: 36.79 ± 0.54 cells per 0.6 mm$^2$) (Fig. 5f, g), thereby demonstrating the specificity of engram-specific CREB-mediated transcription in the consolidation of long-term contextual fear memory. Next, using an independent group (Supplementary Fig. 13a), we confirmed that mice receiving the mCREB virus exhibited no impairments in short-term (5 h) contextual fear memory (Supplementary Fig. 13b) or long-term (72 h) auditory fear memory (Supplementary Fig. 13c), further establishing the specificity of DG engram CREB signaling in the consolidation of contextual fear memory. Finally, WT mice receiving the mCREB virus exhibited no deficits in memory (Supplementary Fig. 12a and Supplementary Fig. 13d), confirming the functional specificity of post-training mCREB expression.

**Discussion**

Elucidating the mechanisms underlying the successful consolidation of memory remains a major goal of neuroscience. Sparse populations of neurons in the DG are known to be critical for the consolidation of long-term memories[2,3]. However, the molecular mechanisms underlying engram-specific consolidation

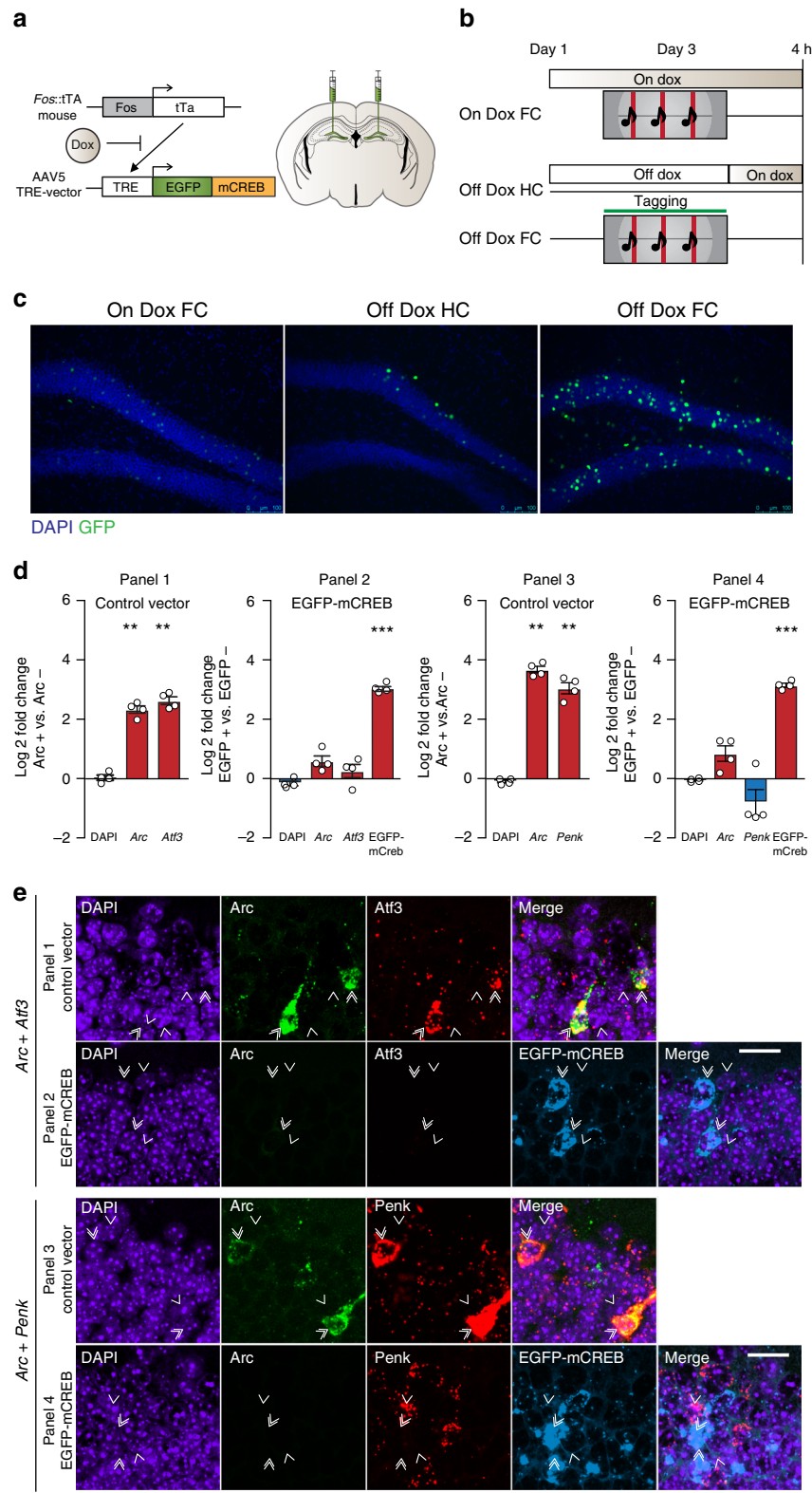

remain largely unknown. Using differential transcriptome profiling of fluorescently tagged DG engram cells and their non-activated neighbors, we revealed genes unique to the consolidation of contextual fear memory. Importantly, using in vivo imaging we established that our activity-dependent *Arc* reporter was persistently expressed within largely the same subset of DG granule cells for at least 24 h following a single-conditioning session, thereby validating our approach for transcriptome profiling during memory consolidation. Furthermore, we also validated the utility of activity-dependent transcriptome profiling by demonstrating the critical requirement of the identified engram-specific changes in CREB-dependent transcription for mediating contextual memory consolidation.

**Fig. 4** Disruption of CREB function prevents regulation of CREB target genes. **a** Experimental design. *Fos*::tTA mice were injected with AAV5-TRE::EGFP-mCREB targeting the DG. **b** On-Dox FC group remained on Dox throughout the experiment, while the off-Dox (HC and FC) groups were placed back on Dox immediately after training. Animals were sacrificed 4 h post-training. **c** Representative images demonstrating expression of EGFP-mCREB in DG neurons after fear conditioning. FC training on Dox induced very low expression of *Fos*::tTa driven EGFP-mCREB. Among animals off Dox, fear-conditioned animals (FC) showed much higher EGFP-mCREB expression than HC controls. Scale bar: 100 μm. **d** Multiplex RNA-scope experiments validate the use of mCREB to disrupt the expression of CREB target genes. Log$_2$ fold change of fluorescence intensity between *Arc*$^+$ and neighboring *Arc*$^-$ cells is reported for the control vector and EGFP$^+$ vs. EGFP$^-$ cells for mCREB injected animals. Analysis of variance: Panel 1 and 2-Control vector ($n = 4$): *Arc*: $F_{(1,7)} = 51.64$, $P = 3.7 \times 10^{-4}$, *Atf3*: $F_{(1,7)} = 22.16$, $P = 3.3 \times 10^{-3}$. EGFP-mCREB vector ($n = 4$): EGFP: $F_{(1,7)} = 79.13$, $P = 1.1 \times 10^{-4}$. Panel 3 and 4-Control vector (n = 4): *Arc*: $F_{(1,7)} = 14.45$, $P = 8.9 \times 10^{-3}$, *Penk*: $F_{(1,7)} = 42.60$, $P = 6.2 \times 10^{-4}$. EGFP-mCREB vector ($n = 4$): EGFP: $F_{(1,7)} = 96.21$, $P = 6.5 \times 10^{-5}$. *$P < 0.05$, **$P < 0.01$, ***$P < 0.001$. Data are presented as mean ± SEM. Source data are provided as a Source Data file. **e** Representative images demonstrating co-expression patterns of *Arc* (green) and *Atf3* (red) or *Arc* (green) and *Penk* (red) in animals injected with the control vector (panels 1 and 3) and EGFP-mCREB (cyan, panels 2 and 4) in the DG of animals injected with the EGFP-mCREB virus. DAPI (blue) labels all cells. Double arrows indicate *Arc*$^+$/*EGFP*$^+$ cells, while single arrows indicate neighboring *Arc*$^-$/*EGFP*$^-$ cells. Scale bar: 20 μm

Memory consolidation is a highly dynamic process requiring multiple waves of gene transcription and protein translation[13,35,36], in order to stabilize and perpetuate experience-dependent changes in synaptic strength and connectivity[37]. The examination of transcriptome changes in activated ensembles has previously been limited to the initial hours following a behavioral experience[9,24,38,39], due to the transient nature of most IEGs used to tag activated neural ensembles. This has limited the identification of key molecular players to the first wave of ARGs that are transcribed rapidly upon stimulation[24,37], while potentially missing out on the identification of downstream gene programs that are specific to synaptic and assembly consolidation as well as to memory persistence. However, the sustained activation of *Arc* in DG engram cells provided us with the opportunity to identify molecular adaptations during memory consolidation 24 h after conditioning, a time-point at which most LTM tests are performed as this is well beyond the window of short-term memory, IEG activation, and vulnerability to protein synthesis inhibition. Moreover, using in vivo imaging we also confirmed that the DG engram neuronal ensemble remained stable throughout the 24 h consolidation period. Using only two principle components, the transcriptome profile of DG engram cells recruited during FC strongly separated from neighboring DG granule cells taken from the same fear conditioned mice, as well as DG granule cells (dVenus$^+$ and dVenus$^-$) from NS and HC groups. Cell types in different brain regions may have vastly different transcriptome profiles[40,41] and only one prior study has looked at gene expression in activated DG cells granule cells, albeit 1 h following novel context exposure[8]. Our findings now significantly expand this approach by determining sustained alterations in gene expression during long-term memory consolidation.

In total, we identified 204 differentially expressed genes in the FC group that surpassed the genome-wide significant threshold of $P < 4.2 \times 10^{-6}$ (Bonferroni correction of $\alpha = 0.05$ for the total $n = 11,802$ genes that passed QC) and validated the co-expression patterns of *Arc*, *Atf3*, *Penk*, and *Kcnq3*, which were identified as among the most significantly regulated genes in DG memory ensemble neurons 24 h after FC. Of these 4 genes, only *Arc* was identified in 4[9,19–21] of the 8 other transcriptome profiling screens we compared our data against (Supplementary Data 4). Given the well-described immediate early response of this gene[42], it is not surprising that 3 of the 4 screens also identified *Arc*, as the animals were sacrificed for RNA extraction within an hour of stimulation. Strikingly, *Penk*, and *Atf3*, genes with known functions in synaptic[43] and structural plasticity[26], were among the most robustly upregulated genes in our screen. Conversely, *Kcnq3*, the most downregulated gene was one of a group of 11 differentially expressed K+ channel genes of which 10 were significantly downregulated, indicative of sustained alterations of DG engram

cell intrinsic excitability during fear memory consolidation, a mechanism that may serve to bind together experiences acquired closely together in time[10,37,44].

An earlier study examined Pavlovian FC in mice with a global homozygous germline deletion of *Atf3*[26]. No differences were observed for contextual FC, while *Atf3*$^{-/-}$ knockout mice showed an enhancement of the strength of auditory FC that is presumably hippocampal-independent. Our findings of a fear memory engram-specific upregulation of *Atf3* following contextual FC, therefore, suggest that germline deletion of *Atf3* is accompanied by homeostatic compensations, at least within the DG. Moreover, these results also offer an important cautionary note regarding the predictive validity of global pretraining molecular genetic deletions compared to region- and engram-specific post-training manipulations as we have performed in the current study.

Transcription factor network analysis revealed that 22 of the top 50 differentially expressed genes were CREB-dependent, including *Arc*, *Atf3*, *Penk*, *Cdkn1a*, *Sorcs3*, and *Inhba*. The transcription factor CREB has previously been implicated in the (1) allocation of neurons to a memory trace through modulation of neuronal excitability[1,6,45,46] as well as (2) memory consolidation. However, most previous studies[33,47–52] that have manipulated CREB function, do so prior to memory acquisition. Moreover, although these studies have indeed demonstrated a critical role for CREB in memory, it has been difficult to ascertain whether the resulting behavioral alterations were due to impairments in allocation, acquisition, consolidation, or some combination thereof. Here, using a *Fos*-driven doxycycline-based inducible system, we were able to repress CREB-mediated transcription for a fixed temporal window during consolidation specifically within the sparse DG engram. Notably, chronic expression of mCREB in the hippocampus was shown to impair memory 7 days after conditioning but not at 24 h[48], indicative of ongoing transcriptional programs that may be specific to memory persistence. However, the mechanisms underlying the function of CREB in engram-specific consolidation and memory persistence has remained thus far largely unknown. Therefore, our findings of an active CREB network at 24 h required for contextual fear memory consolidation firmly establishes the causality of CREB-dependent transcription specifically within the DG engram. Moreover, these results also substantially expand our knowledge of the identity of specific CREB target genes involved in long-term memory.

Taken together, we have identified critical molecular mechanisms that are necessary for the formation of stable memories by sparse DG engram neurons. Moreover, we demonstrate that RNA sequencing in combination with activity-dependent cellular tagging holds considerable promise for elucidating the molecular adaptations following experience-

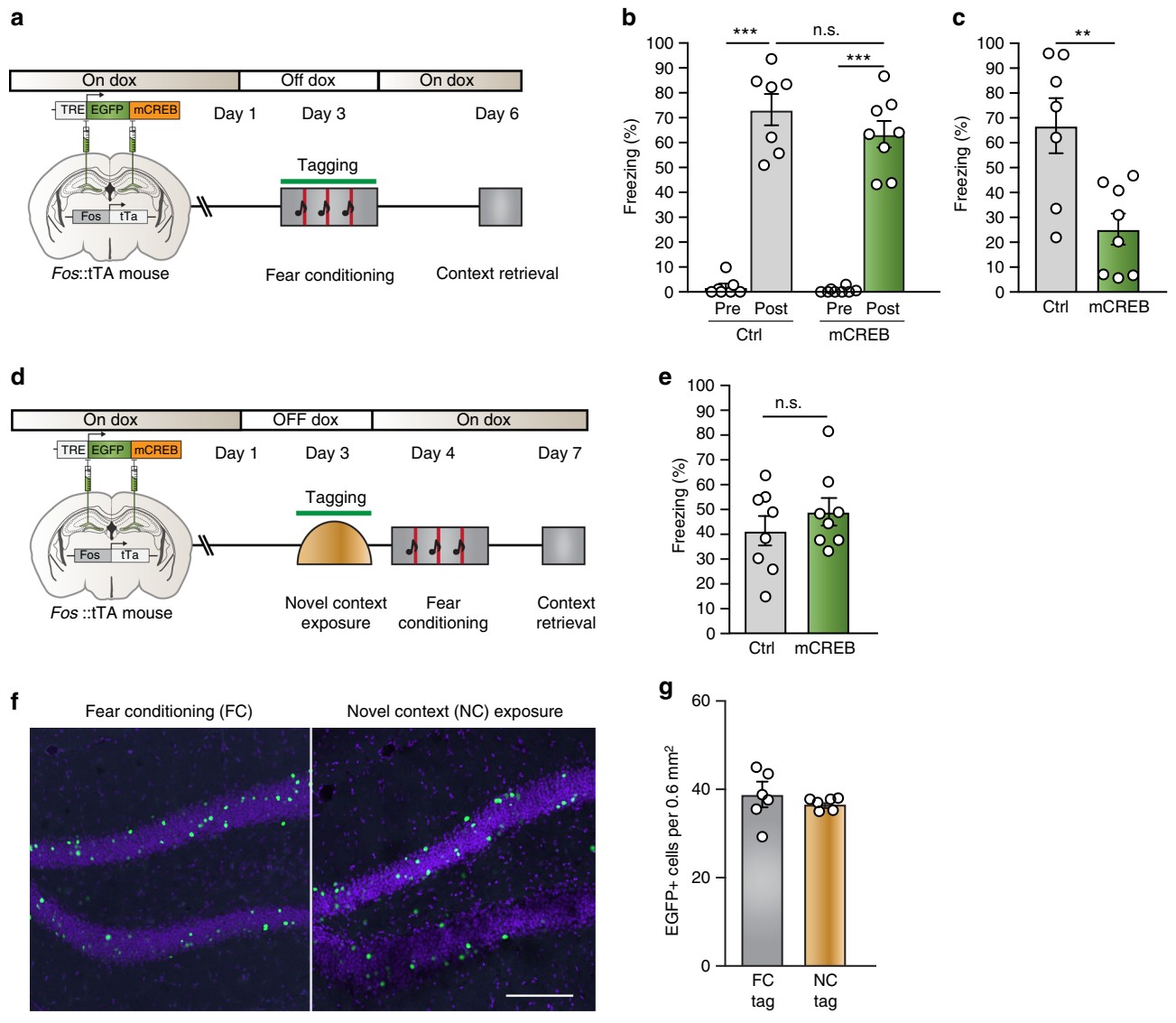

**Fig. 5** DG engram-specific disruption of CREB function impairs memory consolidation. **a** Experimental design. *Fos*::tTA mice were injected with AAV5-TRE:: EGFP-mCREB (*n* = 8) or control AAV5-TRE::mCherry (*n* = 7) targeting the DG, and subsequently taken off Doxycycline prior to fear conditioning. Animals were placed back on Dox immediately after fear conditioning and tested for contextual memory 72 h later. **b** Freezing levels (%) during the training session, prior to footshock onset (pre) and following the termination of the last footshock (post). Analysis of variance: Control Pre vs. Post: $F_{(1,13)}$ = 103.4, $P$ = 3.0 × 10$^{-7}$, mCREB Pre vs. Post: $F_{(1,15)}$ = 163.8, $P$ = 2.2 × 10$^{-9}$, Control vs. mCREB (Post): $F_{(1,14)}$ = 1.4, $P$ = 0.26. **c** Mice injected with mCREB exhibited a significant contextual memory deficit when tested 72 h after training. Analysis of variance $F_{(1,14)}$ = 11.41, $P$ = 0.005. **d** Experimental design. *Fos*::tTA mice were injected with AAV5-TRE::EGFP-mCREB (*n* = 8) or control AAV5-TRE::mCherry (*n* = 8) targeting the DG, and subsequently taken off Doxycycline prior to exposure to a novel context. Animals were placed back on Dox immediately after and fear conditioned 24 h later followed by a contextual memory test 72 h after that. **e** Mice with mCREB expression in cells active during novel context exposure exhibited no memory deficit when tested 72 h after training. **f, g** The same number of DG cells express EGFP-mCREB after exposure to either the fear-conditioning context or a novel context or. **f** Representative images and **g** quantification of the number of EGFP-mCREB cells per 0.6 mm$^2$. Scale bar: 200 μm. n.s. not significant, **$P$ < 0.01, ***$P$ < 0.001. Data are presented as mean ± SEM. Source data are provided as a Source Data file

dependent plasticity with broad applicability throughout the nervous system.

## Methods

**Experimental model and subject details**. Male *Arc*::dVenus and *Fos*::tTa transgenic mice backcrossed more than 10 generations into C57BL/6J were single housed and maintained on a 12 h light/dark cycle with food and water available ad libitum. Experiments were performed during the light phase using adult mice (postnatal weeks 8–12). All experiments were performed in accordance with Dutch law and licensing agreements using protocols ethically approved by the Animal Ethical Committee of the Erasmus MC Rotterdam and Vrije Universiteit Amsterdam.

**Fear conditioning**. Animals explored the conditioning chamber (context A) for 180 s prior to the onset of 3 auditory stimuli (30 s, 5 kHz, 85 dB) that co-terminated with a mild foot shock (0.75 mA, 2 s)[10]. The intertrial interval between tone-shock presentations was 210 s. The conditioning chamber was thoroughly cleaned with 70% ethanol between animals. NS animals underwent the same protocol, but did not receive any foot shocks. HC controls received no exposure to the conditioning chamber and remained in standard housing conditions until they were sacrificed.

Context fear memory retrieval: Animals were exposed to the conditioning context (A) for 180 s at specified time-points after conditioning.

Auditory fear memory retrieval: Animals were exposed to a novel context (B) for 120 s, followed by presentation of the auditory CS for 60 s. This context was thoroughly cleaned with 1% acetic acid between animals and differed in shape, texture, and smell to the conditioning context A.

mCREB experiments: Off Dox-FC animals were taken off food containing doxycycline 48 h prior to conditioning and placed on high-Dox food immediately after. On Dox-FC animals were kept on Dox throughout the experiment. Off Dox-HC animals followed the same Dox schedule as the Off Dox-FC group, but remained in their home cage. For the novel context exposure experiment, animals were taken off Dox food 48 h prior to exposure to a novel context and placed on high-Dox food immediately after and fear-conditioned 24 h later while on Dox.

**Immunohistochemistry.** Animals were deeply anesthetized with Pentobarbitol (50 mg per kg) and perfused with 4% paraformaldehyde (Sigma-Aldrich Chemie N.V., The Netherlands). Brains were dissected and postfixed in 4% paraformaldehyde for 2 h at 4 °C and then transferred to phosphate buffer (0.1 M PB, pH 7.3) containing 10% sucrose and stored overnight at 4 °C. Embedding was performed in 10% gelatin + 10% sucrose, followed by fixation in 30% sucrose containing 10% PFA for 2 h at room temperature. Brains were then immersed in 30% sucrose at 4 °C until slicing. Forty-micrometer coronal sections were collected serially using a freezing microtome (Leica, Wetzlar, Germany; SM 2000R) and stored in 0.1 M PB. Approximately, 15 free floating sections at intervals of 160 μm, across the rostrocaudal axis of the DG were used for immunohistochemistry. For Arc and c-Fos stainings, antigen retrieval was performed at 80 °C for 1 h in 10 mM sodium citrate buffer, prior to pre-incubation with blocking solution (0.1 M PBS) containing 0.5% Triton X-100 (Sigma-Aldrich Chemie N.V., The Netherlands) and 10% normal horse serum (Thermo Fisher Scientific, The Netherlands). Sections were then incubated in primary antibodies (Arc: 1:200, C-7 sc-17839, Santa Cruz, Germany, c-Fos: 1:500, antibody (4): sc-52, Santa Cruz, Germany) for 48–72 h at 4 °C followed by incubation with corresponding Alexa-conjugated secondary antibodies (1:200, Jackson Immunoresearch, Bioconnect, The Netherlands) for 2 h at room temperature. For Atf3 stainings, sections were incubated with the primary antibody (1:100, C-19, sc-188, Santa Cruz, Germany) for 24 h at 4 °C prior to secondary antibody incubation as described above. Both primary and secondary antibodies were diluted in 0.1 M PBS buffer containing 0.4% Triton X-100 and 2% NHS. Nuclear staining was performed using DAPI (300 nmol per l, Thermo Fisher Scientific, The Netherlands) and sections were mounted on slides and coverslipped using Vectashield antifade mounting medium (H-1000, Vector Labs, USA)

**Confocal microscopy and cell counting.** A Zeiss LSM 700 confocal microscope (Zeiss, The Netherlands) was used to make z-stacks of the DG at ×10 or ×20 magnification and 0.5× zoom. Native dVenus, Cy3 or Alexa 555, Alexa647 and DAPI were imaged using the excitation wavelengths of 488, 555, 639, and 405 nm, respectively[10]. The 488, 555, and 639 channels were acquired sequentially so as to avoid bleed-through, and prevent emission spectral overlap. The DAPI channel was acquired in combination with one of the other channels.

For individual counts of dVenus+ cells, 10x images (1.3 mm × 1.3 mm) acquired from the confocal were imported into ImageJ and the Cell counter plugin (V 2.2) was utilized to mark and count dVenus+ cells manually in the granule cell layer of the DG, from 2D projections of the z-stack. The number of Arc-dVenus+ neurons was counted at 160 μm intervals across the entire rostrocaudal axis of the DG using coronal brain sections (Supplementary Fig. 1D). The average number of dVenus+ cells per 1.3 mm × 1.3 mm section in the DG is presented throughout the text[2]. For Atf3+ cell counts, 20x z-stack images (0.6 mm × 0.6 mm) were acquired and counted in the same way as described above.

For colabeling experiments, 20x images were imported to ImageJ where they were digitally merged to form composite images. First, individual cells were marked and counted in separate channels (e.g., native dVenus fluorescence, Arc labeled with Alexa 647 and c-Fos labeled with Cy3). Representative images were edited in ImageJ to generate 2D projections of z-stacks, and all images were treated identically.

**Surgeries.** All surgeries were performed under stereotaxic guidance using co-ordinates from the brain atlas[53] to target the DG (A/P: –1.9, M/L: +/–1, D/V: –2). Isoflurane (1–3% inhalant to effect, up to 5% for induction, RB Pharmaceuticals, UK) was used for general anesthesia and Lidocaine (2%, Sigma-Aldrich Chemie N. V., The Netherlands) provided topical analgesia for all surgeries. Animals received peri-operative analgesia (Temgesic, 0.1 mg per kg, RB Pharmaceuticals, UK) and were closely monitored for postoperative care.

**Microendoscopy.** Implantation of microendoscopes was performed as described in Resendez et al.[54], with minor modifications. Briefly, animals under isoflurane anesthesia were placed on a stereotaxic setup. The skull was cleaned with ethanol (Thermo Fisher Scientific, The Netherlands), Betadine (Gezondheidswinkel VoordeligVitaal, The Netherlands), and hydrogen peroxide (VWR international B.V., The Netherlands) prior the placement of a skull screw (Selva Benelux, The Netherlands). After performing a craniotomy of 1 mm diameter, a column of tissue just above the selected co-ordinates was gently vacuum-aspirated with a 30G blunt needle (SAI Infusion Technologies, USA) and intermittent irrigation using sterile saline. A 1 mm GRIN lens (GLP-1040, Inscopix Inc. USA) was slowly inserted (100–200 μm per min) to ~200 μm above the selected co-ordinates and fixed in place using Vetbond (VWR international B.V., The Netherlands) and dental cement (Contemporary Ortho-Jet Powder & Liquid, Lang Dental Manufacturing, USA). Two weeks after lens implantation, the baseplate (Inscopix Inc., USA) for a miniaturized microscope

(Inscopix Inc., USA) was implanted above the microendoscope lens after determining the best field of view of landmarks like blood vessels and/or DG neurons.

**Viral vectors.** The pAAV-TRE$_{tight}$::EGFP-mCREB plasmid was constructed by replacing hM3Dq-mCherry in pAAV-TRE$_{tight}$::hM3Dq-mCherry (Addgene plasmid #66795, gift from William Wisden) with the coding sequence of EGFP-mCREB from pAAV-mCREB (Addgene plasmid #68551, gift from Eric Nestler)[55] using SLiCE[56]. Viral packaging of pAAV-TRE$_{tight}$::EGFP-mCREB was implemented for AAV2 serotype 5 for in vivo application.

Animals were placed on Doxycline containing food 1 week prior to surgeries[57]. Animals under general isoflurane anesthesia and topical lidocaine anesthesia were placed on a stereotaxic setup and 0.5 μl of virus was bilaterally injected into the selected co-ordinates using a micro-injection pump[58] (CMA 400 syringe pump, Aurora Borealis Control B.V., The Netherlands) at the rate of 0.1 μl per min, followed by an additional 10 min to allow diffusion. The wound was closed with a surgical staple system (Fine Science Tools, Germany) and mice remained in their HC for 3 weeks prior to the start of experiments.

**In vivo imaging of dVenus fluorescence.** Animals implanted with base plates were briefly anesthetized using isoflurane for attachment of miniature microscopes and imaging. The adjusted field of view was briefly imaged an hour prior to FC. The same field of view was then imaged 5 and 24 h after FC for a period of 10 s to minimize photobleaching (Supplementary Fig. 14a). The 5 h time-point was chosen because (1) previous reports have reported maximal experience-driven *Arc*::dVenus expression occurs 4–6 h after stimulation[59,60] and (2) our ex vivo imaging studies demonstrated a consistent proportion of dVenus+ neurons in the DG between 1 and 24 h. Images collected were preprocessed and adjusted to predefined vascular landmarks using the "Name landmarks and register" plugin in ImageJ (V 2.0.0-rc-43/1.50i) (Supplementary Fig. 14b).

**Mouse brain slice preparation for RNA-Seq.** Coronal slices of the hippocampus were prepared from fear conditioned, NS or HC control *Arc*::dVenus mice. Mice were deeply anaesthetized, transcardially perfused, and decapitated before the brain was dissected from the skull. The brain was subsequently mounted and sliced in oxygenated ice-cold slicing medium containing (in mM): N-methyl-D-glucamine 93, KCl 2.5, NaH2PO4 1.2, NaHCO$_3$ 30, HEPES 20, glucose 25, sodium ascorbate 5, sodium pyruvate 3, MgSO$_4$ 7, CaCl$_2$ 0.5, at pH 7.4 adjusted with 10 M HCl. Following the cutting procedure, the slices were maintained on ice in the oxygenated slice medium until the end of the experiment.

**Fluorescence-guided nucleated patch aspiration for RNA-seq.** Individual dentate gyrus granule cells were collected for sequencing using a modified methodology for pulling nucleated patches[14]. Briefly, green (dVenus+) and nongreen cells (dVenus−) were visualized using IR-DIC (Olympus BX51, Olympus Nederland B.V.) on a patch clamp rig constantly perfused with ice-cold slicing medium (temperature in recording chamber was 6 °C). Individual borosilicate glass pipettes (3–4 MΩ) with maximum 5 μL of filtered slicing medium were brought into close proximity of the target cell somata. Identical to whole-cell patch clamp recording techniques, during approach a small voltage step (−5 mV, 500 ms) was used to monitor the formation of a giga-ohm seal after contact using fine pressure control. Once a stable giga-ohm seal formed between the soma and the pipette, the contact patch was broken using a brief suction pulse combined with a brief 500 mV voltage step (100–500 μs via EPC10 HEKA amplifier in whole-cell configuration). Series resistance was not constantly monitored after break-in because low access resistance was not strictly required. After patch opening, a small constant negative pressure (maximum 50 mBar) was applied and slowly increased until the cellular contents could be observed moving into the pipette (or the volume of the cell was observed to decrease). As soon as the cell soma began to shrink in volume, the negative pressure was no longer increased but was maintained until the pipette containing the targeted cell was removed from the holder. Typically the nucleus was clearly visible and began blocking the pipette tip within 45 s of applying constant negative pressure. The recording pipette was then slowly retracted out of the tissue to draw the cell contents out of the slice. During retraction, if the giga-ohm seal was lost, the cell was considered compromised and the pipette and its contents were discarded (10% of cells). Once clear of the slice but still in the bath, the negative pressure in the collection pipette was increased to approximately 100mBar and the pipette quickly cleared of the bath. Upon successful removal, the extreme end tip of the pipette and its contents were immediately broken off into the bottom of an Eppendorf tube containing 3.4 μl of ice-cold lysis buffer with 0.2% Triton X-100 (molecular biology grade, Sigma-Aldrich Chemie N.V., The Netherlands) and RNAse inhibitor. Great care was necessary to break off the tip sufficiently above the waiting lysis buffer mixture to avoid capillary action drawing the reaction medium and any previously collected material back into the broken pipette. The collection tube was spun briefly after each cell was inserted to help assure harvested material (including pipette glass) reached the cold lysis buffer. Any pipette solution remaining in the pipette was not aspirated out of the pipettes to avoid unnecessarily diluting the lysis reaction. Two or three cell pairs (dVenus+/dVenus−) were collected from each slice to minimize the tissue time at

temperatures above 4 °C. Ten pairs (dVenus+/dVenus-) of DG granule cells were collected from each mouse and pooled for each sequencing experiment.

**Sample preparation and RNA sequencing.** Full-length cDNA was generated from 3.4 μl of cell lysate using the Smarter2 protocol[12]. cDNA quality and quantity has been checked on Agilent Bioanalyzer, using the high sensitivity DNA assay prior to amplification and sequencing library preparation (Supplementary Fig. 15). Sequencing libraries were generated from 500 pg of cDNA with Illumina's Nextera XT sampleprep kit (Illumina Inc., USA) and sequenced for single-read 50 bp on Illumina HiSeq2500 using Rapid v2 sequencing chemistry (Illumina Inc., USA). Cells from 16 FC animals, and 4 NS and HC were aspirated and used for library preparation and sequencing. Of these, (1) samples that failed quality control for sample preparation (poor cDNA quality), (2) samples that failed quality control for sequencing (very low percentage alignment), or (3) samples that were excluded from analysis as their paired sample failed quality control, have been listed in Supplementary Data 1. For each library, only the sequenced fragments that yield one unique aligment are included in the expression profile. For each library, the number of detected genes is presented in the Supplementary Data 1. In this table, a gene is considered detected if at least one fragment aligns on it (count ≥ 1). After quality control, for the fear-conditioned group, 3 independent technical replicate experiments were performed ($n = 6$, $n = 4$, and $n = 2$ mice). Sample-to-sample principal component analysis separated samples on state of activation (dVenus+ or dVenus−) and not by experiment (Fig. 2b).

**Transcriptome analysis.** Reads were aligned against the mouse reference genome (mm10) with tophat2 version 2.0.13[61]. Read counts per gene were calculated with htseq-count version 0.6.0[62] using NCBI transcript annotation. Differential expression analysis on raw counts was performed in R[63] using the DESeq2 package[18], in a paired design. Briefly, The DESeq method uses a Negative Binomial (aka Gamma-Poisson) distribution to model the counts per gene/sample, in a generalized linear model. After that, a Wald test was utilized to test for significance of the fitted parameters in the generalized linear model and multiple testing correction was performed using the Benjamini & Hochberg (1995) algorithm. Regularized log counts ($\log_2$ scale, normalized with respect to library size) were used for visualization of data for clustering, box plots and heat maps. GO analysis for molecular function was performed in PANTHER (V 13.1) against a background of the 11,802 genes that passed QC, using a Fisher's exact with FDR multiple test correction[64].

For pathway and upstream regulator analysis[65] using Ingenuity Pathway Analysis (IPA, QIAGEN)[66], a Fisher's exact test (right-tailed), where significance indicates the probability of association of molecules from the dataset with the canonical pathway by random chance alone, was used to calculate an overlap $P$ value corresponding to the probability that the dataset genes were drawn from the same distribution as the genes regulated by a given transcription factor. The activation score ($z$-value) is calculated on the basis of experimentally validated gene regulation by comparing whether an upstream transcription regulator has significantly more "activated" predictions than "inhibited" predictions ($z > 0$) or vice-versa ($z < 0$), where significance implies a rejection of the hypothesis that predictions are random with equal probability. $z$-scores greater than 2 or smaller than −2 are considered significant[65].

**RNA-scope in situ hybridization assay and analysis.** RNA-scope analysis was performed as per the manufacturer's instructions[32] (ACD, Bio-techne Ltd., UK). Briefly, animals were fear conditioned and transcardially perfused with sterile 4% PFA in 1× PBS, 24 h later. After a 24 h period of post-fixation at 4 °C, brains were transferred to 10% sucrose in sterile 1× PBS until they sank. This step was repeated with 20 and 30% sucrose. Brains were then embedded in optimal cutting temperature media (Tissue-Tek, VWR, The Netherlands) and placed in the cryostat at −20 °C for 1 h to equilibrate the tissue. Sections measuring 10 μm from the hippocampus were then mounted on SuperFrost Plus slides (VWR, The Netherlands) and allowed to dry at −20 °C for 2 h. Sections were then processed as per the manufacturer's instructions which included pre-treatment with hydrogen peroxide for 10 min and target retrieval at 99 °C for 5 min and treatment with Protease III for 30 min at 40 °C. Hybridization to probes against Arc (Cat. no. 316911-C3), Atf3 (Cat. no. 426891-C1) and EGFP (also recognizes dVenus, Cat. no. 400281-C2) or Arc and Penk (Cat. no. 318761-C1), or Arc and Kcnq3 (Cat. no. 444261-C1) was carried out at 40 °C for 2 h. HRP Signals against each channel (C1–C3) were then sequentially amplified and developed using TSA Plus fluorophores (Perkin Elmer, The Netherlands) at a dilution of 1:1500, where TSA plus Cy3 (Cat. no. NEL744E001KT) was assigned to C1 probes, TSA plus Cy5 (Cat. no. NEL745E001KT) was assigned to C2 probes and TSA plus Fluorescein (Cat. no. NEL741E001KT) to C3 probes. Positive- (Cat. no. 320881, PolR2A-C1, PPIB-C2, and UBC-C3), and negative-control probes (Cat. no. 320871) were included with every experiment to assess sample RNA quality and optimal permeabilization conditions. Sections were counterstained with DAPI for 30 s, coverslipped with ProLong Gold Antifade Mountant and allowed to dry overnight at RT, in the dark. Four images were taken per section at 60x magnification with each of the four channels being acquired sequentially so as to avoid bleed-through, and prevent emission spectral overlap. Images were exported to ImageJ and background subtraction was performed after which ROIs were drawn around Arc+ or EGFP+ cells

or their non-activated neighboring cells, counterstained with DAPI and the intensity of all four channels was measured in arbitrary units for each cell using the MultiMeasure plugin in ImageJ. Neighboring cells were chosen for analysis to mimic the paired-design used during patch clamp aspiration for RNA-Seq experiments. These intensity measurements were then used to calculate Log2 fold change regulations between neighboring cell pairs for each of the genes/probes measured. Care was taken to ensure that all sections belonging to the same experiment were processed and imaged at the same time. For mCREB experiments, Arc was used to label activated cells in control vector treated animals, while EGFP was used to identify mCREB expressing cells as these cells expressed negligible amounts of Arc.

**Quantification and statistical analysis.** Sample sizes (n), test statistics, degrees of freedom, and P values are noted throughout in the main text and figure legends, and in Supplemental Data 3 for RNA-Seq data. No animals were excluded from the behavioral analysis and no virus misplacements were detected. All statistics were performed using SPSS statistics (V 22, IBM). Univariate analysis of variance (ANOVA) was performed to evaluate significance in the temporal expression profile of dVenus+ cells for both training history (exposure to FC) and time, in confocal fluorescence imaging and RNA-scope analyses. Univariate ANOVA was used in the 24 h experiments detailing dVenus expression in different experimental groups (NS and DS) as well as Arc and dVenus colabeling (HC, NS, and DS). Repeated measure ANOVA was utilized to determine contextual conditioning in the mCREB experiments, while a univariate ANOVA was used to evaluate the effect of treatment at the 72 h retrieval test. Univariate ANOVA was used to evaluate the results of the short-term contextual memory and long-term auditory memory tests. Significant differences were followed up by univariate ANOVA or Fishers least significant difference test when appropriate. The threshold for statistical significance was considered at $\alpha = 0.05$ all experiments, with adjustment for multiple comparisons as specified in the text.

## Data availability

All differentially expressed genes with $P < 0.05$ have been listed in Supplementary Data 3. RNAseq data has been submitted to the National Center for Biotechnology Information Gene Expression Omnibus (NCBI GEO Accession GSE129024).

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

## Acknowledgements

The authors would like to thank Yvonne Gouwenberg and Robbert Zalm (VU University, Amsterdam, The Netherlands) for AAV and mCREB vector construct preparation. This work was supported by the Hersenstichting Nederland Fonds to P.R.-R. and J.J.C.; NWO VENI (016.171.033) to P.R.-R.; NWO VIDI (017.106.384), ZonMw Middelgroot (40-00506-9810026), and ALW Middelgroot (834.12.002) to S.A.K.; NWO VIDI to M.C.v.d.O. (016.168.313); ZonMw TOP grant (40-00812-98-15030) to P.R.-R., M.C.v.d.O. and S.A.K.

## Author contributions

D.S. maintained the *Arc*::dVenus populations. P.R-R. and S.A.K. designed the temporal dVenus expression experiments, co-localization experiments, and RNA-scope experiments and P.R-R. performed them. P.R-R. and S.A.K. designed the in vivo imaging experiments, G.J.M. and R.M.C. provided the expertize regarding the miniature microscopes, and P.R-R. and I.M.M. performed surgeries and imaging. P.R-R., W.F.v.I. and S.A.K. designed the RNA-seq experiments and J.J.C., P.R-R., C.G.B. and M.v.d.H. performed them. P.R-R., C.G.B. and M.v.d.H. analyzed the RNA-seq data. P.R-R.,

M.C.v.d.O. and S.A.K. designed the RNA-scope experiments and P.R-R. performed them. P.R-R., M.C.v.d.O. and S.A.K. designed the mCREB experiments, M.R.M. validated mCREB constructs in vitro and P.R.-R., R.J.v.d.L. and M.C.v.d.O. performed in vivo studies. P.R-R., M.C.v.d.O. and S.A.K. wrote the paper with methodological contributions from J.J.C. and M.v.d.H.

## Additional information

**Competing interests:** The authors declare no competing interests.

