## [Peer Review File · Nature Communications]

Reviewers' Comments:

Reviewer #1:

Remarks to the Author:

The goal of the studies described in this manuscript is to characterize the transcriptomic response of “engram cells”—dentate gyrus granule cells activated by fear conditioning—using RNA sequencing of active and non-active granule cell populations. The authors report that the engram cells show distinct transcriptional programs and that this program is highly dependent on the transcription factor CREB. In all, the experiments are well-designed, the results are robust, and the interpretations are appropriate. These studies provide the most detailed picture to date of the transcriptomic changes associated with memory formation in single granule cell neurons of the DG.

Despite this reviewer’s generally favorable impression of the paper, two weaknesses exist and temper enthusiasm slightly. First, and as mentioned in more detail below, the last experiment (upregulation of a dominant negative CREB protein in DG granule cells activated by fear conditioning, FC) may not be directly related to the earlier experiments and does not demonstrate causality of the earlier transcriptomic changes as argued by the authors. Additional experiments would be needed to strengthen this aspect of the study. Secondly, the main finding—that a CREB-dependent transcriptional program is critical for memory consolidation—has been a dominant view for decades now. I do not mean to be dismissive of the strength of the findings presented, just point to the fact that this potentially lowers the impact of the findings to the field.

Comments:

o The use of miniscope imaging of the dentate gyrus in the Arc::dVenus mouse line provide strong evidence that the contextual fear activated DG ensemble stably expresses Arc for at least 24 hours. This finding provides strong justification for subsequent single cell transcriptional profiling experiments, and is, by itself, an important new finding.

o The finding that Atf3 was the most significant and highly enriched gene in the engram cell population was clear, but a bit confusing. As the authors appropriately mention, Atf3 was shown to function in gating strength of fear memories in hippocampus (Pai et al., 2018) the direction of influence was the opposite. In Pai et al. (2018), Atf3^{-/-} mice showed stronger fear conditioning than wt controls. It is not clear how these two findings are reconciled.

o Figure 3 is not clear to this reviewer. I am not sure what to make of the upper scales of figure 3 (activation z-score, log overlap p value). The panels below seem to speak to the log₂ fold changes per gene by group, but I am not sure about these other measures. The authors should describe this better in the figure legend.

o The last experiment (figure 4), while interesting, does not seem to directly address the central hypothesis of the manuscript. The upregulation of the mCREB in engram cells is presumably not instantaneous, and may take (presumably) hours for the transcription/translation/nuclear translocation/dimerization/promoter binding events to effect its impact on neuronal gene transcription. In the earlier experiments, the endogenous CREB-dependent gene expression can happen quickly (as made evident by the dramatic upregulation of dVenus mediated through the Arc promoter, and rapid phosphorylation of CREB evidenced in many past studies. Is this manipulation related to the transcriptomic events measured in the previous experiments? It is not clear from this one experiment. One potential way to resolve this would be using the single cell approach in mice 24 hours after conditioning. Are the genes identified in the earlier studies disrupted in FC engram cells, or are other transcriptional changes observed?

o The manuscript was well-written and clear, and with the exception noted above, the data were presented in a clear fashion.

Reviewer #2:

Remarks to the Author:

Review of: Engram-specific transcriptome profiling of contextual memory consolidation

In this manuscript, the authors set out to evaluate the transcriptome of "engram" cells in the dentate gyrus of mouse hippocampus to evaluate which genes may mediate consolidation processes. To mark engram cells, the authors used the Venus transgenic mouse line that expresses venus under the endogenous Arc promoter. Unlike other IEGs, Arc expression seems to be sustained for at least 24hrs after fear conditioning in the DG. Engram and non-engram cells were aspirated 24hrs after fear conditioning and RNA sequencing performed. The analysis of the data shows specific genes that are upregulated or altered 24 hours after fear conditioning, including CREB-related genes. The authors then show that dominant negative CREB expressed in FOS-driven DG cells interferes with consolidation of fear conditioning.

While this approach is novel, there are some aspects of the data that require addressing. Moreover, the finding that Arc expression is sustained confirms previous work by Bramham et al (e.g. <https://www.ncbi.nlm.nih.gov/pubmed/17898216>) and the role of CREB in consolidation is very well documented in the literature. Therefore, it's disappointing that the authors do not follow up on novel genes or pathways that would illuminate new biology of consolidation processes nor is it clear why the causal experiments were done using a FOS promoter line rather than an Arc as is used to determine gene expression changes.

Major concerns

1. If the venus transgenic Arc mouse is a faithful reporter of endogenous Arc expression, it's unclear why the percentage overlap in Figure 1h is so low in the HC/NS groups or why the overlap would change with behavior. Since the expression of venus in HC/NS mice is "uncoupled" to endogenous Arc expression, it is unclear how to interpret the RNA-seq data from these groups.
3. It is unclear if endogenous Arc protein expression is sustained since only one time point (24hrs) was measured. The authors should look at endogenous Arc expression in venus mice at similar time points as in 1a/b.
4. While supplementary figure 2 shows nice overlap of endogenous Arc and Fos expression 90 min after fear conditioning in the DG, the key experiment of showing venus expression and Fos needs to be carried out.
5. Since the interpretation of all the results in the paper rely on the faithfulness of this transgenic line, it is imperative that these cells are clearly marking "engrams". Yet, the authors use a different transgenic mouse using the FOS promoter to validate the causal role of CREB in DG "engram" cells. To be consistent with the RNA-seq results these experiments should have been done using the Arc promoter, such as developed by Christine Denny (Denney et al, Neuron 2014).

Minor concerns

1. It is very strange to put P values in an abstract.
2. Supplementary figures 10 and 11 are not mentioned in the text and 10 seems to be associated with data in figure 1d

Reviewer #3:

Remarks to the Author:

General:

The authors address the question of the molecular mechanisms underlying memory consolidation by characterizing the transcriptional changes following contextual fear conditioning learning in engram cells. The question is of great interest to the neuroscience community and more generally, characterization of transcriptional changes induced by neuronal activity or learning-and-memory is highly interesting and there is a great need to address it. The authors combine complex tools to label, track and profile RNA in Engram cells, which provide an interesting view into these rare network of cells. Specifically, they label engram cells using the Arc immediate early gene, followed by RNA-seq to unbiasedly characterize the transcriptional changes in these specific rare cells 24 hours following fear conditioning. This method enables unbiased characterization of a rare population of cells but with relatively low throughput, and the total number of cells/samples analyzed in this work seems to be especially low, which is a limitation of the study. In addition – the pooled RNA-seq samples are enriched for Engram cells but not depleted of other cells. Despite this low number of cells, the transcriptional changes seem to be robust, and point to the involvement of CREB, which they functionally validate. Overall, the characterization of a transcriptional network that is activated and necessary for memory consolidation is novel and highlights multiple interesting genes/proteins, however the specific role of CREB in memory consolidation has been shown before and its novelty needs to be clarified.

Specific comments:

Labeling Engram cells and time points: While evidence for the robust labeling of Engram cells was presented, it was shown that there are many labeled cells in the HC and NS (30% in HC compared to FC). When pooled together this implies that 30% of cells collected are not part of the Engram network of cells but will contribute to the expression differences observed. I appreciate that all the tests were compared across the HC and NS conditions to partially address this issue. A more detailed follow-up validating the transcriptional results in the single cell level is required. This could be done using the SmartSeq2 protocol (which is designed for single cells), but also a more small-scale method would be acceptable such as qPCR or ISH. Should ideally show an expression of more than one gene per cell to validate their co-expression patterns.

Time point: As you show, a significant number of cells are labeled Arc+ at earlier time points starting from 1 hour post labeling. The cells at earlier time points should have a dramatically different expression pattern, which will be very informative to compare to the 24 hours profile.

To address this point, the minimum requirement will be to validate the expression of some of the target genes such as Atf3 and of CREB at earlier time points, which is expected to reveal the dynamics of the process, and also will help establish CREB as the master regulator orchestrating this response. Please also clarify your rationale for focusing on the 24 hours time point in the main text.

Number of samples: The total number of samples analyzed for RNA-seq (the total libraries collected and the ones passing quality filter), in each of the conditions, should be clearly stated in the main text and methods.

Variability between libraries: The variability between the libraries across the conditions should be addressed and presented, especially due to the low number of cells pooled in each sample. This is relevant for: (1) generally across all genes – can be presented as a correlation matrix displaying the correlation between each pair of samples. (2) Specifically for the differentially expressed genes – the distribution of the expression of the differential genes should be presented (as a heatmap or other visualization). Currently you only present the average expression values per condition in Figure 3 and S5.

Quality controls: There is no information in the main text and the supplement regarding the quality controls for the RNA-seq libraries. Did all the libraries pass a basic quality filter (including for example the number of genes detected, number of reads mapped, and the percent of mapping to the genome and the transcriptome)? What are the criteria for excluding genes? The information regarding these quality tests and filters, including the number of reads, genes and mapping rates of each library should be provided in the methods section and the main text. The only number provided is the total number of genes included in the analysis across all cells.

Statistics: Was there a multiple hypothesis correction method used in every statistical test (please specify the specific test and thresholds used) for the differential expression test of genes, enrichment of pathways and transcription factor targets. The threshold mentioned is 0.05 p-value, which according to my understanding is before the correction. In the main text there is no need to mention both the p-value and the corrected value (the corrected/adjusted value is enough).

Transcriptional networks: Please clarify in the methods how the prediction of CREB as a master transcription regulator was done – (1) Was it done using the same statistical test for enrichment of pathways? (2) How are the targets predicted and what is the supporting evidence for these target genes, including the direct and the indirect targets (as noted in Figure 3a). (3) were there any other factors that were significant?

Collection of single cells for RNA-seq: I appreciate the careful planning of the RNA-seq experiments, and especially the efforts to minimize the time of the cells at 4c. Please clarify how long and what temperature where the cells kept in until the collection was finished.

Comparison to previous work: Several other works measured transcriptional changes and IEGs following other stimuli or at different time points after contextual fear conditioning learning. A direct comparison of the profiles you detect (in FC, HC and NS) is required. This will be used as a validation of your results, will clarify what's unique to FC and clarify what's unique to the 24 hours time point of the FC. When doing this comparison, you can use RNA-seq of pooled samples as well as single cell and nucleus RNA-seq. In addition, also standard single nucleus RNA-seq experiments of total brain tissues (not selected for activation) you can find such signature of activation also in the hippocampus brain region (even if they are not reported as part of their main figures).

IEGs: Based on current RNA-seq studies it is clear that there are many IEGs we haven't identified previously and that their expression patterns differ across different stimuli and cell types. Here you mention that only *Npas4* is detected in the FC response set, however going over the genes in Figure 3, I can detect other genes that their expression is also established as being induced by neuronal activity including: *Penk* and *Bdnf*. To clarify this point, please conduct a more comprehensive analysis of IEGs and comparison to the literature and previous published work.

PCA: The results of the PCA analysis are very strong, however I have the following questions and requests:

- Please specify the overlap between the top genes contributing to PC 1 and PC 2 and the differentially expressed genes reported.

- What are the top genes that separate between the Arc+/- and the FC/HC/NS conditions?

- Why are you using only the top 100 genes in the PCA analysis? Usually the top genes includes many "house keeping" genes that might be subjected to technical variability. Does the analysis work when you use all genes or pre-choose variable genes only?

- Since PCA is a dimension reduction method and not a clustering method - I not use the word "clustering" for separating between samples by PC scores – but rather say that the score distinguishes between two populations of cells or that "PC-X scores separated cells by Y".

Novelty of results: This point is a major point of weakness in the study. It's currently unclear what did you learn about the mechanism of memory consolidation compared to previous experiments, specifically related to CREB's role and requirement for memory consolidation? You should relate to previous evidence showing that the role of DG in the formation and consolidation of new memories, involves many transcription factors, of which CREB is a well-documented one. Moreover, neuronal

population which contains relatively high level of CREB at the time of the learning have been proposed as candidate for being selected as Engram cells.

Transcription network: One aspect of the novelty in this work is the detailed transcriptional network identified and characterized. However, validations of the transcriptional network and its involvement in memory consolidation is required in order to claim that this is indeed novel finding. Specifically, I find the expression of Atf3 especially exciting, since it is part of the CREB/ATF transcription factor family. Another candidate that is exciting is the Penk gene, since its function in the Hippocampus is not well understood compared to other brain regions. To show some functional relevance you would need to first validate the expression in the relevant cells using an alternative method such as ISH, and follow that with an additional experiment, which ideally would be a functional assay but could also be an indirect measurement such as showing that the expression of these target genes is reduced in the CREB knock-out mouse.

Modified methodology for pulling nucleated patches:

- Please specify if your method can work for single cell RNA-seq?
- How does it compare to the Patch-Seq method by Fuzik et al.? (and please add the missing reference to that method)
- The bio-analyzer trace is very low (even compared to single cell RNA-seq)? How can you explain it and how many PCR cycles are you using?

Additional comments:

- Citations: Several other works of single cell or nucleus RNA-seq showing activity induced expression changes are published other than Lacar, et al. 2016. Including for example Sathyamurthy et al. 2018, Hrvatin et al 2018 and Ye et al. 2017.
- Figure 1: barplots can be smaller, and microscopic images larger – so we can actually see the data. Specifically in 1i and 1h
- Fold change is more informative to report in log scale (e.g. you report a 670-fold upregulation of Atf3?)
- Please provide the list of PC 1 and 2 genes as a supplement
- Please provide information regarding the genes within the enriched differential pathways (supplementary table)

Manuscript NCOMMS-18-08965. Response to the reviewers

The authors would like to thank the editor and the reviewers for their positive feedback and insightful comments on how to strengthen the findings of this manuscript.

Based on the reviewers' comments, we have added the following to the revised manuscript:

1. Co-localization between Fos and dVenus expression 90 min after conditioning, in addition to the co-localization between Fos and endogenous Arc (**Supplementary Fig. 2**), further confirming the validity of the dVenus reporter.
2. Expression of endogenous *Arc* at 3 more time-points after fear conditioning (1 h, 5 h and 14 h) in addition to 24 h, thereby strengthening our conclusion that dentate gyrus engram cells exhibit a sustained temporal *Arc* expression profile during the first 24 h of fear memory consolidation (**Supplementary Fig. 4**).
3. Differential expression of a more comprehensive panel of activity-regulated genes (ARGs) at 24 h after fear conditioning (**Fig. 2b**).
4. Multiplex RNAscope validation of top differentially expressed genes identified by RNAseq. Specifically, we have used RNAscope to visualize and quantify the single-cell (co)-expression patterns of up-regulated (*Arc*, *Atf3* and *Penk*) and down-regulated (*Kcnq3*) genes in DG engram cells at 24 h after fear conditioning (**Fig. 3c and 3d**).
5. Provided the additionally requested details regarding the implementation, quality control and analysis of the RNAseq experiments (**Methods**).
6. Added a new table with alignment and expression statistics of all samples, including those that did not pass quality control filters of library preparation or sequencing (**Supplementary Table 1**).
7. Added a new table that lists the individual genes contributing to the principal component analysis (**Supplementary Table 2**).
8. Added a principal component analysis of the top 500 differentially expressed genes, in addition to the PCA plot of the top 100 genes. (**Supplementary Fig. 6**).
9. Provided a sample correlation matrix for all genes to address the variability between libraries (**Supplementary Fig. 8a**).
10. Added heat maps to visualize the distribution of expression of differentially expressed genes for all experimental groups (**Supplementary Fig. 8b-d**).
11. Quantified the single-cell temporal profile of Atf3 protein expression in DG engram cells at 1 h, 5 h, 14 h and 24 h after fear conditioning (**Supplementary Fig. 9**).
12. Compared our RNA-sequencing differential gene expression profiles to previously reported transcriptional changes in the hippocampus or activated neuronal ensembles following various stimuli and/or time-points after fear conditioning (**Supplementary Table 4**).
13. Provided a new table with a list of the individual genes contributing to each of the identified enriched pathways (**Supplementary Table 5**).
14. Provided a list of upstream transcriptional regulators that exhibit a significant overlap *P*-value (**Supplementary Table 6**).
15. Used RNAscope fluorescent *in situ* hybridization to validate the CREB dependence of *Arc*, *Atf3* and *Penk* expression in DG engram cells by engram-specific dominant negative CREB (mCREB) (**Fig. 4d and 4e**).
16. Added an experiment in which we express mCREB in DG cells tagged in a novel context prior to fear conditioning, to further corroborate our findings that CREB mediated transcription specifically in FC engram cells is essential for contextual fear memory consolidation (**Fig. 4d and 4e**).
17. Revised the Discussion section to include the points suggested by the reviewers, including the novelty of our data and the role of CREB in memory consolidation.

In the revised manuscript text, all changes have been indicated in blue.

Please find below a point-by-point reply (in black) to the reviewers' comments (in red):

Reviewer #1 (Remarks to the Author):

The goal of the studies described in this manuscript is to characterize the transcriptomic response of “engram cells”—dentate gyrus granule cells activated by fear conditioning—using RNA sequencing of active and non-active granule cell populations. The authors report that the engram cells show distinct transcriptional programs and that this program is highly dependent on the transcription factor CREB. In all, the experiments are well-designed, the results are robust, and the interpretations are appropriate. These studies provide the most detailed picture to date of the transcriptomic changes associated with memory formation in single granule cell neurons of the DG.

Despite this reviewer's generally favorable impression of the paper, two weaknesses exist and temper enthusiasm slightly.

First, and as mentioned in more detail below, the last experiment (upregulation of a dominant negative CREB protein in DG granule cells activated by fear conditioning, FC) may not be directly related to the earlier experiments and does not demonstrate causality of the earlier transcriptomic changes as argued by the authors. Additional experiments would be needed to strengthen this aspect of the study.

We appreciate the reviewer's endorsement of our work and the importance of the findings. With regard to the conclusiveness of the referenced last experiment, we agree with the reviewer that additional evidence would be needed to strengthen the validity of this result. Therefore, we have now performed multiple additional experiments to address this point. First, we have implemented multiplex RNAscope fluorescent *in situ* hybridization experiments to directly visualize *Arc*, *Atf3* and *Penk* RNA at 24 h after fear conditioning in DG cells with engram-specific expression of mCREB (**Fig. 4d and 4e**). Second, we have also demonstrated a disruption of *Atf3* at the protein level in animals injected with the mCREB vector, further validating the functional relevance of the changes in RNA. Third, we have included an experiment in which we express mCREB in DG cells tagged in a novel context prior to fear conditioning, which further corroborate our findings that CREB mediated transcription specifically in FC engram cells is essential for memory consolidation (**Fig. 4d and 4e**). Taken together, we feel that this series of additional experiments has greatly strengthened the conclusiveness of the requirement for CREB mediated transcription and the identification of novel engram-specific genes regulated during memory consolidation.

Secondly, the main finding—that a CREB-dependent transcriptional program is critical for memory consolidation—has been a dominant view for decades now. I do not mean to be dismissive of the strength of the findings presented, just point to the fact that this potentially lowers the impact of the findings to the field.

We thank the reviewer for bringing up this important point. While we agree that the general concept of a CREB-dependent transcriptional program being necessary for consolidation is not new, our study provides a significant step forward in uncovering the identity and quantitative regulation of CREB-dependent genes during memory consolidation, a major unanswered question of long-standing interest to the field. Moreover, our findings also demonstrate that the experience-dependent CREB transcriptional program remains persistently active 24 h after fear conditioning, a temporal window that is notably longer during consolidation than has ever previously been studied due to the technical limitations of relying exclusively upon other IEGs (such as *Fos*) commonly used to tag and capture engram

cells, which we have now been able to overcome by taking advantage of the sustained transcriptional activity of *Arc*.

Lastly, we would like to emphasize that to the best of our knowledge, this is the first study to perform a region- and engram-restricted disruption of CREB mediated transcription specifically during consolidation. In contrast, the vast majority of work in the field of contextual fear memory has implemented more global manipulations of CREB function (**Table A**), leading to difficulty in disentangling cell-autonomous effects from those that are due to network-level perturbations. Additionally, given the well-documented role of CREB in memory allocation, pre-training disruptions of CREB function – the most frequently implemented approach to interrogating the necessity of CREB function in studies of cognition (**Table A**) – make it difficult to ascertain whether the observed behavioral effects result from impaired memory allocation, acquisition and/or consolidation. Accordingly, the high spatiotemporal precision of our manipulations in the present study has allowed for a unique opportunity to reduce or eliminate these historical confounds.

Taken together, we hope that we have convinced the reviewer not only of the novelty of our findings but also their potential impact in substantially furthering our knowledge regarding the identity of key molecular players involved in the long-term consolidation of memory by engram cells.

Reference	Timing of manipulation	Brain region	Type of manipulation	Behavior effect
Bourtchuladze et al, Cell 1994	From birth	Whole brain	CREB $\alpha\Delta$ mutant mice	LTM deficit
Gass et al, Learning & Memory 1998	From birth	Whole brain	CREB $\alpha\Delta$ isoform and CREB _{comp} mice	Gene dosage-dependent LTM deficit
Rammes et al., European Journal of Neuroscience 2001	From birth	Forebrain	Dominant negative (CREB _{S133A}) mice	No contextual LTM deficit
Graves et al., Hippocampus 2002	From birth	Whole brain	CREB $\alpha\Delta$ isoform mutant mice on B6/129 F1 hybrid background	STM and LTM deficit
Kida et al, Nature Neuroscience 2002	6 h prior to conditioning	Whole brain	Repression of CREB transcription using the tamoxifen inducible CREB ^{IR} transgenic system	LTM deficit
Trifilieff et al, Learning & Memory 2006	1 h or 9 h after conditioning	Hippocampus (CA1)	Pharmacological disruption of both ERK1/2 and CREB pathways	LTM deficit
Peters et al, Genes Brain & Behavior 2009	3 days prior to conditioning	Hippocampus (CA1)	siRNA against CREB	LTM deficit
Viosca et al, Learning & Memory 2009	1 week prior to conditioning	Whole brain	Constitutively active CREB protein in VP16-CREB mutant mice	Formation of protein synthesis resistant LTM
Suzuki et al, Journal of Neuroscience 2011	From birth	Whole brain	Gain of function: Transgenic mice expressing dominant active CREB mutants	Enhanced STM and LTM

Kathirvelu et al, Neurobiology of Learning and Memory 2013	Three weeks prior to conditioning	Dorsal hippocampus	Repression of CREB transcription by the use of mCREB	7 day LTM deficit
Serita et al, Scientific Reports 2017	From birth	Forebrain	Constitutive activation of CREB, dominant active mutant of CREB (DIEDML mice)	Enhanced LTM in TFC

Table A. Examples of CREB manipulations that affect long-term contextual fear memory

Comments:

o The use of miniscope imaging of the dentate gyrus in the Arc::dVenus mouse line provide strong evidence that the contextual fear activated DG ensemble stably expresses Arc for at least 24 hours. This finding provides strong justification for subsequent single cell transcriptional profiling experiments, and is, by itself, an important new finding.

We appreciate the reviewer's endorsement of the importance of this finding.

o The finding that *Atf3* was the most significant and highly enriched gene in the engram cell population was clear, but a bit confusing. As the authors appropriately mention, *Atf3* was shown to function in gating strength of fear memories in hippocampus (Pai et al., 2018) the direction of influence was the opposite. In Pai et al. (2018), *Atf3*^{-/-} mice showed stronger fear conditioning than wt controls. It is not clear how these two findings are reconciled.

We agree with the reviewer that the results described in the Pai *et al.* study¹ may appear counterintuitive to our findings. However, it is important to note that this study of germline *Atf3*^{-/-} knockout mice found **no** differences in contextual fear memory (using a similar protocol to ours). In contrast, the enhancement of fear conditioning observed by Pai *et al.* was specific to auditory fear freezing, which is notably hippocampus independent. Furthermore, we would also like to add that a global germline knockout of *Atf3* might result in homeostatic compensatory changes that influence fear memory acquisition in a manner distinct from a region- and engram-specific post-training manipulation^{2, 3, 4} as we have performed in the current manuscript. Accordingly, we have now modified the main text to reflect this distinction.

o Figure 3 is not clear to this reviewer. I am not sure what to make of the upper scales of figure 3 (activation z-score, log overlap p value). The panels below seem to speak to the log2 fold changes per gene by group, but I am not sure about these other measures. The authors should describe this better in the figure legend.

We apologize for the limited description, and have now updated the figure legend as suggested by the reviewer to better define these metrics, which were implemented as defined by Kramer et al⁵. In brief, activation Z-scores are calculated from cross-correlations of gene regulation to identify putative co-regulated genes and calculated per experimental group using both magnitude and direction (up- versus down-regulation). For a given regulator *r*, the overlap *p*-value *p*(*r*) estimates the probability of finding a similar or higher number of *r*-regulated genes by random chance.

o The last experiment (figure 4), while interesting, does not seem to directly address the central hypothesis of the manuscript. The upregulation of the mCREB in engram cells is presumably not instantaneous, and may take (presumably) hours for the transcription/translation/nuclear translocation/dimerization/promoter binding events to effect its impact on neuronal gene transcription. In the earlier experiments, the endogenous CREB-

dependent gene expression can happen quickly (as made evident by the dramatic upregulation of dVenus mediated through the Arc promoter, and rapid phosphorylation of CREB evidenced in many past studies. Is this manipulation related to the transcriptomic events measured in the previous experiments? It is not clear from this one experiment. One potential way to resolve this would be using the single cell approach in mice 24 hours after conditioning. Are the genes identified in the earlier studies disrupted in FC engram cells, or are other transcriptional changes observed?

We thank the reviewer for this suggestion. In order to address this point, we have performed multiplex RNAscope labeling to evaluate the expression of 3 identified CREB network targets (*Arc*, *Atf3* and *Penk*) in mCREB expressing FC engram cells at 24 h after fear conditioning (**Figure 4**). Furthermore, we also confirm that the mCREB-mediated disruption of *Atf3* RNA expression at 24 hours is paralleled by an absence of *Atf3* protein expression (**Supplementary Figure 9**). Together, these additional data further substantiate the causality of the engram cell-specific mCREB manipulation. While we agree that it would be very interesting to perform a genome-wide transcriptional analysis of engram cells following the mCREB manipulation, we hope the reviewer agrees that such an extensive additional analysis is beyond the scope of the current study.

o The manuscript was well-written and clear, and with the exception noted above, the data were presented in a clear fashion.

We sincerely appreciate the reviewer's support and helpful suggestions, which we feel have further strengthened the manuscript.

Reviewer #2 (Remarks to the Author):

In this manuscript, the authors set out to evaluate the transcriptome of “engram” cells in the dentate gyrus of mouse hippocampus to evaluate which genes may mediate consolidation processes. To mark engram cells, the authors used the Venus transgenic mouse line that expresses venus under the endogenous Arc promoter. Unlike other IEGs, Arc expression seems to be sustained for at least 24hrs after fear conditioning in the DG. Engram and non-engram cells were aspirated 24hrs after fear conditioning and RNA sequencing performed. The analysis of the data shows specific genes that are upregulated or altered 24 hours after fear conditioning, including CREB-related genes. The authors then show that dominant negative CREB expressed in FOS-driven DG cells interferes with consolidation of fear conditioning.

While this approach is novel, there are some aspects of the data that require addressing. Moreover, the finding that Arc expression is sustained confirms previous work by Bramham et al (e.g. <https://www.ncbi.nlm.nih.gov/pubmed/17898216>) and the role of CREB in consolidation is very well documented in the literature. Therefore, it's disappointing that the authors do not follow up on novel genes or pathways that would illuminate new biology of consolidation processes nor is it clear why the causal experiments were done using a FOS promoter line rather than an Arc as is used to determine gene expression changes.

We thank the reviewer for these insightful comments and would like to point out the following:

1) The study mentioned by the reviewer (Messaoudi *et al.*⁶) examined the expression of *Arc* for up to 4 hours following the induction of medial perforant path-to-dentate gyrus LTP in anesthetized rats. In contrast, our findings substantially expand upon this observation by demonstrating that contextual fear conditioning results in an engram-specific expression of *Arc* that is sustained during memory consolidation for at least 24 h.

2) Our *in vivo* miniscope imaging data establish that *Arc*⁺ engram cells are a stable population defined early in the consolidation process, an important novel finding given that previous studies^{6, 7} have been limited to temporal profiling of *Arc* expression by *in situ* hybridization using a cross-sectional experimental design.

3) While we agree that the general concept of a CREB-dependent transcriptional program being necessary for consolidation is not new, our study provides a significant step forward in uncovering the identity and quantitative regulation of CREB-dependent genes during memory consolidation, a major unanswered question of long-standing interest to the field. Moreover, our findings also demonstrate that an experience-dependent CREB transcriptional program remains persistently active 24 h after fear conditioning, a temporal window that is notably deeper into consolidation than has ever previously been studied due to the technical limitations of relying exclusively upon other IEGs (such as *Fos*) commonly used to tag and capture engram cells, which we have now been able to overcome by taking advantage of the sustained transcriptional activity of *Arc*.

Moreover, we would like to emphasize that to the best of our knowledge, this is the first study to perform a region- and engram-restricted disruption of CREB mediated transcription specifically during consolidation. In contrast, the vast majority of work in the field of contextual fear memory has implemented more global manipulations of CREB function (**Table A**), leading to difficulty in disentangling cell-autonomous effects from those that are due to network-level perturbations. Additionally, given the well-documented role of CREB in memory allocation, pre-training disruptions of CREB function – the most frequently implemented approach to interrogating the necessity of CREB function in studies of cognition (**Table A**) –

make it difficult to ascertain whether the observed behavioral effects result from impaired memory allocation, acquisition and/or consolidation. Accordingly, the high spatiotemporal precision of our manipulations in the present study has allowed for a unique opportunity to reduce or eliminate these historical confounds.

Finally, in addition to the CREB network, we also identify a number of other differentially expressed gene clusters (e.g., voltage-gated potassium channels) that may be critical to mechanisms that underlie the stabilization of neuronal intrinsic and synaptic alterations needed for successful memory encoding. In this regard, we have committed to making our RNAseq data freely available through the GEO repository upon publication of this manuscript for other scientists to independently explore.

Taken together, we hope that we have convinced the reviewer not only of the novelty of our findings but also their potential impact in substantially furthering our knowledge regarding the identity of key molecular players involved in the long-term consolidation of memory by engram cells.

Reference	Timing of manipulation	Brain region	Type of manipulation	Behavior effect
Bourtchuladze et al, Cell 1994	From birth	Whole brain	CREB $\alpha\Delta$ mutant mice	LTM deficit
Gass et al, Learning & Memory 1998	From birth	Whole brain	CREB $\alpha\Delta$ isoform and CREBcomp mice	Gene dosage-dependent LTM deficit
Rammes et al., European Journal of Neuroscience 2001	From birth	Forebrain	Dominant negative (CREB _{S133A}) mice	No contextual LTM deficit
Graves et al., Hippocampus 2002	From birth	Whole brain	CREB $\alpha\Delta$ isoform mutant mice on B6/129 F1 hybrid background	STM and LTM deficit
Kida et al, Nature Neuroscience 2002	6 h prior to conditioning	Whole brain	Repression of CREB transcription using the tamoxifen inducible CREB ^{IR} transgenic system	LTM deficit
Trifilieff et al, Learning & Memory 2006	1 h or 9 h after conditioning	Hippocampus (CA1)	Pharmacological disruption of both ERK1/2 and CREB pathways	LTM deficit
Peters et al, Genes Brain & Behavior 2009	3 days prior to conditioning	Hippocampus (CA1)	siRNA against CREB	LTM deficit
Viosca et al, Learning & Memory 2009	1 week prior to conditioning	Whole brain	Constitutively active CREB protein in VP16-CREB mutant mice	Formation of protein synthesis resistant LTM
Suzuki et al, Journal of Neuroscience 2011	From birth	Whole brain	Gain of function: Transgenic mice expressing dominant active CREB mutants	Enhanced STM and LTM
Kathirvelu et al, Neurobiology of Learning and Memory 2013	Three weeks prior to conditioning	Dorsal hippocampus	Repression of CREB transcription by the use of mCREB	7 day LTM deficit

Serita et al, Scientific Reports 2017	From birth	Forebrain	Constitutive activation of CREB, dominant active mutant of CREB (DIEDML mice)	Enhanced LTM in TFC
--	------------	-----------	---	---------------------

Table A. Examples of CREB manipulations that affect long term contextual fear memory

Major concerns

1. If the venus transgenic Arc mouse is a faithful reporter of endogenous Arc expression, it's unclear why the percentage overlap in Figure 1h is so low in the HC/NS groups or why the overlap would change with behavior. Since the expression of venus in HC/NS mice is "uncoupled" to endogenous Arc expression, it is unclear how to interpret the RNA-seq data from these groups.

We appreciate the opportunity to clarify this important point. As demonstrated previously and re-confirmed in the current manuscript, the dVenus transgenic Arc mouse line has been extensively validated as a faithful reporter of endogenous Arc transcriptional activity^{8, 9}. However, it is important to note that the half-life of endogenous Arc protein (~30 min) is significantly shorter than dVenus (~3h)^{8, 10, 11}. Therefore, in the fear conditioned (FC) group which exhibits sustained activation of Arc transcription, the co-expression of endogenous Arc protein and dVenus protein is very high. In contrast, in the HC/NS groups which have only transient epochs of Arc transcription, the reduced half-life of endogenous Arc protein compared to dVenus protein results in a substantial fraction of endoArc(-) / dVenus(+) cells, compared to double positive cells. The HC/NS groups therefore allowed an ideal within-subject experiment design, in which the dVenus(+) cells from these groups were recently activated but not as members of a contextual fear memory engram. Consequently, differentially expressed genes from the HC/NS groups could be seen as putatively representing transcriptomic responses to non-specific behavioral experience, compared to acquisition of a contextual fear memory.

2. It is unclear if endogenous Arc protein expression is sustained since only one time point (24hrs) was measured. The authors should look at endogenous Arc expression in venus mice at similar times points as in 1a/b.

We thank the reviewer for suggesting this experiment and have now also measured endogenous Arc expression at 1 h, 5 h and 14 h after fear conditioning (in addition to 90 min and 24 h) (**Supplementary Figure 4**) to conclusively demonstrate the sustained profile of endogenous Arc expression in DG engram cells.

3. While supplementary figure 2 shows nice overlap of endogenous Arc and Fos expression 90 min after fear conditioning in the DG, the key experiment of showing venus expression and Fos needs to be carried out.

We agree and have now added this to **Supplementary Figure 2**.

4. Since the interpretation of all the results in the paper rely on the faithfulness of this transgenic line, it is imperative that these cells are clearly marking "engrams". Yet, the authors use a different transgenic mouse using the FOS promoter to validate the causal role of CREB in DG "engram" cells. To be consistent with the RNA-seq results these experiments should have been done using the Arc promoter, such as developed by Christine Denny (Denney et al, Neuron 2014).

We have chosen to utilize the *Fos* promoter-driven line for conducting the manipulation studies following the *Arc*⁺ engram cell-specific RNAseq on the basis of the following criteria: a) The co-localization of *Arc* and *Fos* is very high in DG granule cells following contextual fear conditioning (**Supplementary Figure 2**), b) the *Fos* promoter-driven line that we used has been extensively characterized and validated^{12, 13, 14, 15, 16}, and c) the use of *Fos* instead of *Arc* provided a more stringent criteria for behavioral validation of the functional relevance of the RNAseq findings by employing an independent promoter to confirm that the manipulations are specific to engram cells while avoiding potentially confounding effects of using the *Arc* promoter for driving behavioral manipulations due to the sustained transcriptional activity and consequent reduction in temporal specificity of transgenic activation at the time of conditioning (see for example the comparison of the *Fos* and *Arc* promoters using the Targeted Recombination in Active Populations (TRAP) technology)¹⁷. Moreover, as demonstrated in the Denny *et al.* Neuron 2014 study¹⁸ cited by the reviewer, the TRAP approach has relatively slow temporal kinetics, requiring at least 36 hours post-activation for tamoxifen activation, cre recombination, and asymptotic transgene expression, which would have been problematic for investigating the functional relevance of the RNAseq analyses performed at 24 hours post-conditioning.

Minor concerns

1. It is very strange to put P values in an abstract.

We apologize for any misunderstanding and are willing to remove the p-values from the abstract at the discretion of the editor.

2. Supplementary figures 10 and 11 are not mentioned in the text and 10 seems to be associated with data in figure 1d.

Supplementary Figures 10 and 11 (now re-numbered as **Supplementary Figures 14 and 15** in the revised manuscript) are mentioned in the Methods section of the text.

Reviewer #3 (Remarks to the Author):

General:

The authors address the question of the molecular mechanisms underlying memory consolidation by characterizing the transcriptional changes following contextual fear conditioning learning in engram cells. The question is of great interest to the neuroscience community and more generally, characterization of transcriptional changes induced by neuronal activity or learning-and-memory is highly interesting and there is a great need to address it. The authors combine complex tools to label, track and profile RNA in Engram cells, which provide an interesting view into these rare network of cells. Specifically, they label engram cells using the Arc immediate early gene, followed by RNA-seq to unbiasedly characterize the transcriptional changes in these specific rare cells 24 hours following fear conditioning. This method enables unbiased characterization of a rare population of cells but with relatively low throughput, and the total number of cells/samples analyzed in this work seems to be especially low, which is a limitation of the study. In addition – the pooled RNA-seq samples are enriched for Engram cells but not depleted of other cells. Despite this low number of cells, the transcriptional changes seem to be robust, and point to the involvement of CREB, which they functionally validate. Overall, the characterization of a transcriptional network that is activated and necessary for memory consolidation is novel and highlights multiple interesting genes/proteins, however the specific role of CREB in memory consolidation has been shown before and its novelty needs to be clarified.

We thank the reviewer for the endorsement of our work.

Specific comments:

Labeling Engram cells and time points:

While evidence for the robust labeling of Engram cells was presented, it was shown that there are many labeled cells in the HC and NS (30% in HC compared to FC). When pooled together this implies that 30% of cells collected are not part of the Engram network of cells but will contribute to the expression differences observed. I appreciate that all the tests were compared across the HC and NS conditions to partially address this issue. A more detailed follow-up validating the transcriptional results in the single cell level is required. This could be done using the SmartSeq2 protocol (which is designed for single cells), but also a more small-scale method would be acceptable such as qPCR or ISH. Should ideally show an expression of more than one gene per cell to validate their co-expression patterns.

We thank the reviewer for this comment and have now used multiplex RNAscope technology to validate the expression changes observed in engram cells after fear conditioning. Specifically, using RNAscope we have now validated the expression of four genes (up-regulated: *Arc*, *Atf3* and *Penk*; down-regulated: *Kcnq3*) at the single-cell level *in situ*. Furthermore, these data also confirm the co-expression of *Atf3*, *Penk*, and *Kcnq3* with *Arc* at the single-cell level. We have also employed a probe for GFP to further validate that the expression changes observed are specific to dVenus+ cells 24 h after fear conditioning. **(Figure 3)**.

Lastly, we would also like to point out that cells activated in home cage conditions may indeed be the cells that are recruited to a fear memory engram after a learning experience, as shown by previous work from our laboratory using the same *Arc::dVenus* reporter mouse in the lateral amygdala⁸. Therefore, it is quite possible that the 30% of labeled cells in HC compared to FC may not represent non-specific activation outside the engram, but rather *bona fide* engram cells in the FC condition.

Time point: As you show, a significant number of cells are labeled Arc+ at earlier time points starting from 1 hour post labeling. The cells at earlier time points should have a dramatically different expression pattern, which will be very informative to compare to the 24 hours profile.

To address this point, the minimum requirement will be to validate the expression of some of the target genes such as Atf3 and of CREB at earlier time points, which is expected to reveal the dynamics of the process, and also will help establish CREB as the master regulator orchestrating this response. Please also clarify your rationale for focusing on the 24 hours time point in the main text.

We appreciate the opportunity to clarify this very important point. The goal of our study was to investigate the molecular mechanisms of memory consolidation. There is widespread agreement within the field that 24 hours post-training is firmly within the window of long-term memory consolidation. Moreover, the 24h time point is well beyond the window of short-term memory, immediate early gene activation, and vulnerability to protein synthesis inhibition. Lastly, from a study design perspective, experiments performed 24h post-conditioning avoid potential confounding effects of differences in circadian regulation. Therefore, we hope the reviewer would agree that performing RNAseq at earlier time points post-conditioning, although interesting, are beyond the aim of our current study.

As suggested by the reviewer, we have examined the temporal expression of Atf3 using a well-validated antibody (C-19, sc-188, Santa Cruz) at 1h, 5h, 14h and 24h post-conditioning. Importantly, we found that nearly every Atf3+ cell was also dVenus+ (**Supplementary Figure 9**), as initially suggested by the RNAseq analyses and now further demonstrated at the single-cell level at 24h using RNAscope (**Figure 3**). Moreover, consistent with the reviewer's prediction of distinct genes being differentially expressed during early vs. later time points, we also observed a bimodal distribution in the temporal profile of Atf3+ cells, with peaks at 5 h and 24 h post-conditioning. Lastly, and confirmatory of the fear conditioning-induced CREB dependent up-regulation of Atf3 expression, engram-specific post-conditioning expression of mCREB abolished the increase of Atf3+ cells (**Supplementary Figure 9**).

The reviewer also suggested examining the temporal profile of pCREB, however as has been the experience of many investigators in the field, the available pCREB antibodies suffer from very poor immunohistochemical labeling that is inadequate for single-cell quantification. We used the most frequently cited pCREB antibody (#06-519, Merck Millipore)^{19, 20} (**Figure A**). However, the combination of the newly included single-cell data by temporal profiling of Atf3 and engram-specific blockade of *Atf3* expression by mCREB, along with multiplex target validation using RNAscope, together establishes strong evidence in support of our conclusion that CREB-mediated transcription is necessary for the long-term consolidation of contextual fear memories by dentate gyrus engram cells.

Figure A. pCREB antibody labeling in the dentate gyrus. Non-specific staining of dentate gyrus granule cells with the pCREB antibody after fear conditioning. Scale bar: 100µm

Number of samples:

The total number of samples analyzed for RNA-seq (the total libraries collected and the ones passing quality filter), in each of the conditions, should be clearly stated in the main text and methods.

We fully agree. This information has now been added to the main text and methods.

Variability between libraries:

The variability between the libraries across the conditions should be addressed and presented, especially due to the low number of cells pooled in each sample. This is relevant for:

(1) generally across all genes – can be presented as a correlation matrix displaying the correlation between each pair of samples.

We have now included a sample-to-sample correlation plot (**Supplementary Figure 8**) to demonstrate the correlation between the different samples.

(2) Specifically for the differentially expressed genes – the distribution of the expression of the differential genes should be presented (as a heatmap or other visualization).

Currently you only present the average expression values per condition in Figure 3 and S5.

We have now included heat maps in **Supplementary Figure 8** to visualize the distribution of differentially expressed genes for the home-cage (HC), no-shock (NS) and fear-conditioned (FC) groups.

Quality controls:

There is no information in the main text and the supplement regarding the quality controls for the RNA-seq libraries. Did all the libraries pass a basic quality filter (including for example the number of genes detected, number of reads mapped, and the percent of mapping tot the genome and the transcriptome)?

What is the criteria for excluding genes? The information regarding these quality tests and filters, including the number of reads, genes and mapping rates of each library should be provided in the methods section and the main text. The only number provided is the total number of genes included in the analysis across all cells.

We have now included a new table (**Supplementary Table 1**) with alignment and expression statistics. In this table, we clarify that some samples failed sample prep QC (primarily on the basis of cDNA quality), sequencing QC (very low percentage alignment), or because the corresponding within-subject paired dVenus(+)/dVenus(-) sample failed. For each library, sequenced fragments that yielded only one unique alignment were included in the expression profile. A gene was considered detectable if at least one fragment could be aligned to it (count ≥ 1).

Statistics:

Was there a multiple hypothesis correction method used in every statistical test (please specify the specific test and thresholds used) for the differential expression test of genes, enrichment of pathways and transcription factor targets. The threshold mentioned is 0.05 p-value, which according to my understanding is before the correction. In the main text there is no need to mention both the p-value and the corrected value (the corrected/adjusted value is enough).

Yes, multiple testing correction was performed using the Benjamini & Hochberg (1995) algorithm. We implemented the DESeq method²¹, which utilizes a Negative Binomial (Gamma-Poisson) distribution to model counts per gene/sample in a generalized linear model. Wald's test was used for statistical testing of the fitted parameters in the generalized linear model.

As suggested by the reviewer, we have now reported only the adjusted p-value in the main text.

Transcriptional networks:

Please clarify in the methods how the prediction of CREB as a master transcription regulator was done –

(1) Was it done using the same statistical test for enrichment of pathways?

The pathway and upstream regulator analyses were performed using Ingenuity Pathway Analysis (IPA):

a) Significance values for the canonical pathways were calculated using Fisher's Exact Test (right-tailed) in which significance indicates the probability of association of molecules from the dataset with the canonical pathway by random chance. We utilized a significance threshold of $P < 0.01$ for the IPA analyses.

b) Upstream Regulator Analysis (URA) is based on expected causal effects between upstream regulators and their downstream targets on the basis of manual curation from the published literature. URA examines the targets of each upstream regulator, compares the observed direction of target change with the expected direction, and calculates a prediction

for each upstream regulator. The direction of change is the gene expression in the experimental samples relative to a control. If the direction of change is:

- Consistent with the literature: IPA predicts that the upstream regulator is more active in the experimental sample than in the control.
- Explicitly inconsistent with the literature (i.e., anti-correlated with the literature): IPA predicts that the upstream regulator is less active in the experimental sample than in the control.
- Not clear (there is a random pattern relative to the literature): IPA does not make an activation or inhibition prediction for the upstream regulator. However, there may still be a significant overlap (Fisher's Exact p-value), albeit without a clear pattern of directionality for confidently predicting activation or inhibition.

Four parameters are used to calculate the p-value for each regulator using Fisher's Exact Test:

- 1) Genes known to be regulated by the regulator (i.e., connected downstream of a regulator using E, T, or PD edges) AND are in the dataset.
- 2) Genes known to be regulated by the regulator BUT NOT in the dataset.
- 3) Genes NOT regulated by the regulator BUT ARE in the dataset.
- 4) Genes curated as being downstream of any regulator BUT NOT in the dataset and NOT regulated.

(2) How are the targets predicted and what is the supporting evidence for these target genes, including the direct and the indirect targets (as noted in Figure 3a)

Targets are predicted using IPA-based causal analysis⁵, utilizing a structured collection of nearly 5 million experimental findings manually curated from the biomedical literature or integrated from third-party databases. The network contains 40,000 nodes that represent mammalian genes and pharmacological compounds of known function. Nodes are connected by 1,480,000 edges representing experimentally observed cause-effect relationships that relate to expression, transcription, activity, molecular modification, inter-molecular binding, and transport, including the direction of the causal effect (i.e., activating or inhibiting).

Using engram-specific post-training induction of mCREB with RNAscope, we have now validated that 3 of the identified target genes (*Arc*, *Atf3* and *Penk*) are downstream of CREB. To the best of our knowledge, this is the first study performed *in vivo* to validate multiple putative CREB targets at the single-cell level.

(3) Where there any other factors that were significant?

Yes, several upstream regulators were identified with significant overlap *P* values. However, the CREB1 regulator pathway was exclusively predicted as being activated and encompassed the most genes. The full list has now been added as **Supplementary Table 6**.

Collection of single cells for RNA-seq:

I appreciate the careful planning of the RNA-seq experiments, and especially the efforts to minimize the time of the cells at 4c. Please clarify how long and what temperature where the cells kept in until the collection was finished.

We thank the reviewer for the endorsement of our experimental design. We have now clarified in the Methods that cells were maintained at 1°C – 4°C during sample collection, with a strict requirement for sequential collection of 10 dVenus⁺/dVenus⁻ pairs of neighboring DG granule cells per mouse during a maximum of 1-2 hours.

Comparison to previous work:

Several other works measured transcriptional changes and IEGs following other stimuli or at different time points after contextual fear conditioning learning. A direct comparison of the profiles you detect (in FC, HC and NS) is required. This will be used as a validation of your results, will clarify what's unique to FC and clarify what's unique to the 24 hours time point of the FC. When doing this comparison, you can use RNA-seq of pooled samples as well as single cell and nucleus RNA-seq. In addition, also standard single nucleus RNA-seq experiments of total brain tissues (not selected for activation) you can find such signature of activation also in the hippocampus brain region (even if they are not reported as part of their main figures).

We have now compared our differential gene expression data sets (FC, NS and HC) to 8 different data sets available from 4 independent studies:

1. Cho J, *et al.* Multiple repressive mechanisms in the hippocampus during memory formation. *Science* **350**, 82-87 (2015): **4 data sets**
2. Cho JH, Huang BS, Gray JM. RNA sequencing from neural ensembles activated during fear conditioning in the mouse temporal association cortex. *Sci Rep* **6**, 31753 (2016): **1 data set**
3. Hermeijer G, Mahlke C, Gutzmann JJ, Schreiber J, Bluthgen N, Kuhl D. Genome-wide profiling of the activity-dependent hippocampal transcriptome. *PLoS One* **8**, e76903 (2013): **2 data sets**
4. Lacar B, *et al.* Nuclear RNA-seq of single neurons reveals molecular signatures of activation. *Nat Commun* **7**, 11022 (2016): **1 data set**

The results are provided in **Supplementary Table 4**. Consistent with cell-type and brain region specificity, and experience-dependent transcriptional regulation, although a degree of overlap is observed between our data set and the aforementioned ones, the majority of differentially expressed genes from our data set appear unique to 1) DG engram cells, and 2) 24 h after fear conditioning.

IEGs:

Based on current RNA-seq studies it is clear that there are many IEGs we haven't identified previously and that their expression patterns differ across different stimuli and cell types. Here you mention that only *Npas4* is detected in the FC response set, however going over the genes in Figure 3, I can detect other genes that their expression is also established as being induced by neuronal activity including: *Penk* and *Bdnf*. To clarify this point, please conduct a more comprehensive analysis of IEGs and comparison to the literature and previous published work.

We have now included a more comprehensive list in **Figure 2** that details a substantial number of activity regulated genes that are differently expressed 24 h after fear conditioning. As suggested by the reviewer, this list was compiled in comparison to previously published work.

PCA: The results of the PCA analysis are very strong, however I have the following questions and requests:

- Please specify the overlap between the top genes contributing to PC 1 and PC 2 and the differentially expressed genes reported.
- What are the top genes that separate between the Arc+/- and the FC/HC/NS conditions?

PC1 and PC2 correspond to the genes that separate between the Arc+/- and the FC/HC/NS conditions, respectively. In the analysis of the top 100 genes, PC1 separates between Arc+/- conditions; while in the analyses of the top 500 genes and of all 11,802 genes, PC1 separates between FC/HC/NS conditions. Here, we provide the list of top genes that separate between the Arc+/- and the FC/HC/NS conditions, with their square loadings. In bold, we have indicated which genes are differentially expressed ($P_{adj} < 0.01$, \log_2 fold change > 1.0).

Top genes separating between the Arc+/- condition with their squared loadings

Gene name	Squared loadings
Arc	0.07799230
Atf3	0.05634442
Inhba	0.05515490
Blnk	0.04972477
Sorcs3	0.03928676
Acan	0.03470173
Penk	0.03342708
Gpnmb	0.03307988
Nptx2	0.03187137
C1qa	0.02659418

Top genes separating between the FC/HC/NS conditions with their squared loadings

Gene name	Squared loadings
Rgl2	0.05697525
Lrrc45	0.04465957
Dalrd3	0.03994531
Snx29	0.03530563
Gja1	0.03045594
Guf1	0.02835739
Ppap2a	0.02646135
Zfp956	0.02505679
Zfp692	0.02358926
Sstr2	0.02275560

- Why are you using only the top 100 genes in the PCA analysis? Usually the top genes includes many “house keeping” genes that might be subjected to technical variability. Does the analysis work when you use all genes or pre-choose variable genes only?

Yes, the analysis looks very similar when more genes are included. We have now included a PCA analysis of the top 500 genes as **Supplementary Figure 6**. In addition, **Figure B** demonstrates that a PCA analysis of all 11,802 genes results in a similar pattern of separation between samples.

Figure B. Sample-to-sample principal component analysis of all 11,802 genes that passed QC.

- Since PCA is a dimension reduction method and not a clustering method - I not use the word “clustering” for separating between samples by PC scores – but rather say that the score distinguishes between two populations of cells or that “PC-X scores separated cells by Y”.

We fully agree and have changed the wording in the main text accordingly.

Novelty of results:

This point is a major point of weakness in the study. It's currently unclear what did you learn about the mechanism of memory consolidation compared to previous experiments, specifically related to CREB's role and requirement for memory consolidation?

You should relate to previous evidence showing that the role of DG in the formation and consolidation of new memories, involves many transcription factors, of which CREB is a well-documented one. Moreover, neuronal population which contains relatively high level of CREB at the time of the learning have been proposed as candidate for being selected as Engram cells.

We thank the reviewer for bringing up this important point. While we agree that the general concept of a CREB-dependent transcriptional program being necessary for consolidation is not new, our study provides a significant step forward in uncovering the identity and quantitative regulation of CREB-dependent genes during memory consolidation, a major unanswered question of long-standing interest to the field. Moreover, our findings also demonstrate that an experience-dependent CREB transcriptional program remains persistently active 24 h after fear conditioning, a temporal window that is notably deeper into consolidation than has ever previously been studied due to the technical limitations of relying exclusively upon other IEGs (such as *Fos*) frequently used to tag and capture engram cells, which we have now been able to overcome by taking advantage of the sustained transcriptional activity of *Arc*. Furthermore, Our *in vivo* miniscope imaging data establish that *Arc*⁺ engram cells are a stable population defined early in the consolidation process, an important novel finding given that previous studies^{6, 7} have been limited to temporal profiling of *Arc* expression by *in situ* hybridization using a cross-sectional experimental design. In addition, with regard to earlier hypotheses proposing that relatively high levels of CREB at the time of learning may be a candidate mechanism for engram selection, we would also like to point out that dVenus⁺ cells activated in home cage conditions may indeed be the cells that are recruited to a fear memory engram after a learning experience, as shown by previous work from our laboratory using the same *Arc* reporter mouse in the amygdala⁸ and comprehensively summarized in our recent review article on the topic²².

Lastly, we would like to emphasize that to the best of our knowledge, this is the first study to perform a region- and engram-restricted disruption of CREB mediated transcription specifically during consolidation. In contrast, the vast majority of work in the field of contextual fear memory has implemented more global manipulations of CREB function (**Table A**), leading to difficulty in disentangling cell-autonomous effects from those that are due to network-level perturbations. Additionally, given the well-documented role of CREB in memory allocation, pre-training disruptions of CREB function – the most frequently implemented approach to interrogating the necessity of CREB function in studies of cognition (**Table A**) – make it difficult to ascertain whether the observed behavioral effects result from impaired memory allocation, acquisition and/or consolidation. Accordingly, the high spatiotemporal precision of our manipulations in the present study has allowed for a unique opportunity to reduce or eliminate many of these historical confounds.

Taken together, we hope that we have convinced the reviewer not only of the novelty of our findings but also their potential impact in substantially furthering our knowledge regarding the identity of key molecular players involved in the long-term consolidation of memory by engram cells.

Reference	Timing of manipulation	Brain region	Type of manipulation	Behavior effect
Bourtchuladze et al, Cell 1994	From birth	Whole brain	CREB Δ mutant mice	LTM deficit
Gass et al, Learning & Memory 1998	From birth	Whole brain	CREB Δ isoform and CREB ^{comp} mice	Gene dosage-dependent LTM deficit
Rammes et al., European Journal of Neuroscience 2001	From birth	Forebrain	Dominant negative (CREB _{S133A}) mice	No contextual LTM deficit
Graves et al., Hippocampus 2002	From birth	Whole brain	CREB Δ isoform mutant mice on B6/129 F1 hybrid background	STM and LTM deficit
Kida et al, Nature Neuroscience 2002	6 h prior to conditioning	Whole brain	Repression of CREB transcription using the tamoxifen inducible CREB ^{IR} transgenic system	LTM deficit
Trifilieff et al, Learning & Memory 2006	1 h or 9 h after conditioning	Hippocampus (CA1)	Pharmacological disruption of both ERK1/2 and CREB pathways	LTM deficit
Peters et al, Genes Brain & Behavior 2009	3 days prior to conditioning	Hippocampus (CA1)	siRNA against CREB	LTM deficit
Viosca et al, Learning & Memory 2009	1 week prior to conditioning	Whole brain	Constitutively active CREB protein in VP16-CREB mutant mice	Formation of protein synthesis resistant LTM
Suzuki et al, Journal of Neuroscience 2011	From birth	Whole brain	Gain of function: Transgenic mice expressing dominant active CREB mutants	Enhanced STM and LTM
Kathirvelu et al, Neurobiology of Learning and Memory 2013	Three weeks prior to conditioning	Dorsal hippocampus	Repression of CREB transcription by the use of mCREB	7 day LTM deficit
Serita et al, Scientific Reports 2017	From birth	Forebrain	Constitutive activation of CREB, dominant active mutant of CREB (DIEDML mice)	Enhanced LTM in TFC

Table A. Examples of CREB manipulations that affect long term contextual fear memory

Transcription network:

One aspect of the novelty in this work is the detailed transcriptional network identified and characterized. However, validations of the transcriptional network and its involvement in memory consolidation is required in order to claim that this is indeed novel finding. Specifically, I find the expression of Atf3 especially exciting, since it is part of the CREB/ATF transcription factor family. Another candidate that is exciting is the Penk gene, since its

function in the Hippocampus is not well understood compared to other brain regions. To show some functional relevance you would need to first validate the expression in the relevant cells using an alternative method such as ISH, and follow that with an additional experiment, which ideally would be a functional assay but could also be an indirect measurement such as showing that the expression of these target genes is reduced in the CREB knock-out mouse.

We thank the reviewer for this excellent suggestion. Therefore, we have now performed multiple additional experiments to address this point. First, we have used multiplex RNAscope fluorescent *in situ* hybridization to validate the expression of four genes [up-regulated: *Arc*, *Atf3* and *Penk*; down-regulated: *Kcnq3*) at the single-cell level *in situ* (**Figure 3**). Second, we have employed a probe for GFP/Venus to further validate that the expression changes observed are specific to dVenus+ cells 24 h after fear conditioning. Third, we have also used RNAscope to directly visualize *Arc*, *Atf3* and *Penk* RNA at 24 h after fear conditioning in DG cells with engram-specific expression of mCREB (**Fig. 4d and 4e**). This experiment further establishes that up-regulation of these genes observed by RNAseq is indeed abolished in the presence of mCREB (**Figure 4**). Fourth, we have additionally confirmed the CREB-dependence of the fear conditioning-induced up-regulation of *Atf3* expression at the protein level by demonstrating that engram-specific post-conditioning expression of mCREB abolishes the increase of *Atf3*+ cells by confocal immunohistochemistry (**Supplementary Fig. 9**). Taken together, we feel that this series of additional experiments has greatly strengthened the conclusiveness of our study.

Modified methodology for pulling nucleated patches:

- Please specify if your method can work for single cell RNA-seq?

We have not tested this, but speculate that it would be possible using the SmartSeq2 protocol. However the quality of RNA generated will depend greatly on the care taken during patch clamp aspiration and RNA isolation, amplification and cDNA synthesis.

- How does it compare to the Patch-Seq method by Fuzik et al.? (and please add the missing reference to that method)

We thank the reviewer for this suggestion and have indeed now added this reference to the manuscript. However, it is difficult to compare our protocol with the Patch-Seq method because:

- 1) the protocol requires harvesting of the cells for sequencing after patch-clamp recordings to measure the electrophysiological properties of the cells, while we aspirate the cellular contents immediately. Thus, our aspiration protocol is much shorter than Fuzik et al.
- 2) we apply less negative pressure (max 50mBar) compared to Fuzik et al (-50mPa) for aspirating the cells.
- 3) the two protocols use different methods for full-length cDNA synthesis, library generation and sequencing.
- 4) the two protocols use different methods for read processing and molecule counts.

- The bio-analyzer trace is very low (even compared to single cell RNA-seq)? How can you explain it and how many PCR cycles are you using?

The low level of the bio-analyzer trace shown in **Supplementary Figure 10** (currently **Supplementary Figure 15**) results from the 6-fold dilution required for the HiSense measurement. The cDNA synthesis was performed using 18 PCR cycles.

Additional comments:

- Citations: Several other works of single cell or nucleus RNA-seq showing activity induced expression changes are published other than Lacar, et al. 2016. Including for example Sathyamurthy et al. 2018. Hrvatin et al 2018 and Ye et al. 2017.

We thank the reviewer for pointing this out and have now included these references.

- Figure 1: barplots can be smaller, and microscopic images larger – so we can actually see the data. Specifically in 1i and 1h

We have now modified **Figure 1h-i** to improve the visualization of the images.

- Fold change is more informative to report in log scale (e.g. you report a 670-fold upregulation of Atf3?)

We have now consistently reported log-fold regulation throughout the manuscript. However, in a few instance for emphasis in the main text, we have reported both the fold regulation as well as the log-fold regulation.

- Please provide the list of PC 1 and 2 genes as a supplement

We have now included these lists as **Supplementary Table 2**.

- Please provide information regarding the genes within the enriched differential pathways (supplementary table)

We have now added this as **Supplementary Table 5**.

References

1. Pai CS, *et al.* The Activating Transcription Factor 3 (Atf3) Homozygous Knockout Mice Exhibit Enhanced Conditioned Fear and Down Regulation of Hippocampal GELSOLIN. *Front Mol Neurosci* **11**, 37 (2018).
2. Han JH, *et al.* Selective erasure of a fear memory. *Science* **323**, 1492-1496 (2009).
3. Sekeres MJ, Neve RL, Frankland PW, Josselyn SA. Dorsal hippocampal CREB is both necessary and sufficient for spatial memory. *Learn Mem* **17**, 280-283 (2010).
4. Yiu AP, *et al.* Neurons are recruited to a memory trace based on relative neuronal excitability immediately before training. *Neuron* **83**, 722-735 (2014).
5. Kramer A, Green J, Pollard J, Jr., Tugendreich S. Causal analysis approaches in Ingenuity Pathway Analysis. *Bioinformatics* **30**, 523-530 (2014).
6. Messaoudi E, *et al.* Sustained Arc/Arg3.1 synthesis controls long-term potentiation consolidation through regulation of local actin polymerization in the dentate gyrus in vivo. *J Neurosci* **27**, 10445-10455 (2007).
7. Ramirez-Amaya V, Angulo-Perkins A, Chawla MK, Barnes CA, Rosi S. Sustained transcription of the immediate early gene Arc in the dentate gyrus after spatial exploration. *J Neurosci* **33**, 1631-1639 (2013).
8. Gouty-Colomer LA, *et al.* Arc expression identifies the lateral amygdala fear memory trace. *Mol Psychiatry* **21**, 1153 (2016).
9. Rudinskiy N, *et al.* Orchestrated experience-driven Arc responses are disrupted in a mouse model of Alzheimer's disease. *Nat Neurosci* **15**, 1422-1429 (2012).
10. Eguchi M, Yamaguchi S. In vivo and in vitro visualization of gene expression dynamics over extensive areas of the brain. *Neuroimage* **44**, 1274-1283 (2009).
11. Soule J, Alme M, Myrum C, Schubert M, Kanhema T, Bramham CR. Balancing Arc synthesis, mRNA decay, and proteasomal degradation: maximal protein expression triggered by rapid eye movement sleep-like bursts of muscarinic cholinergic receptor stimulation. *J Biol Chem* **287**, 22354-22366 (2012).
12. Garner AR, *et al.* Generation of a synthetic memory trace. *Science* **335**, 1513-1516 (2012).
13. Liu X, *et al.* Optogenetic stimulation of a hippocampal engram activates fear memory recall. *Nature* **484**, 381-385 (2012).
14. Ramirez S, *et al.* Creating a false memory in the hippocampus. *Science* **341**, 387-391 (2013).
15. Redondo RL, Kim J, Arons AL, Ramirez S, Liu X, Tonegawa S. Bidirectional switch of the valence associated with a hippocampal contextual memory engram. *Nature* **513**, 426-430 (2014).
16. Reijmers LG, Perkins BL, Matsuo N, Mayford M. Localization of a stable neural correlate of associative memory. *Science* **317**, 1230-1233 (2007).
17. Guenther CJ, Miyamichi K, Yang HH, Heller HC, Luo L. Permanent genetic access to transiently active neurons via TRAP: targeted recombination in active populations. *Neuron* **78**, 773-784 (2013).
18. Denny CA, *et al.* Hippocampal memory traces are differentially modulated by experience, time, and adult neurogenesis. *Neuron* **83**, 189-201 (2014).
19. Han JH, *et al.* Neuronal competition and selection during memory formation. *Science* **316**, 457-460 (2007).
20. Zhang H, Kyzar EJ, Bohnsack JP, Kokare DM, Teppen T, Pandey SC. Adolescent alcohol exposure epigenetically regulates CREB signaling in the adult amygdala. *Scientific reports* **8**, 10376 (2018).

21. Love MI, Huber W, Anders S. Moderated estimation of fold change and dispersion for RNA-seq data with DESeq2. *Genome Biol* **15**, 550 (2014).
22. Rao-Ruiz P, Yu J, Kushner SA, Josselyn SA. Neuronal competition: microcircuit mechanisms define the sparsity of the engram. *Curr Opin Neurobiol* **54**, 163-170 (2018).

Reviewers' Comments:

Reviewer #1:

None

Reviewer #2:

Remarks to the Author:

The authors have added substantial new data that has significantly improved the original manuscript.

The essential controls to validate the authors' conclusions are now included and alleviate my initial concerns.

While I still feel there isn't much novelty of establishing the CREB transcriptional in engram consolidation, I do take the authors' points that the sustained time window of 24 hours is a surprising result and the genes identified in the data set can be followed up on by the field.

I just had a minor point to clarify on the ATF3 protein data set in supplemental figure 9. It's surprising that the protein expression is so low (~4x less expression than Venus) give that the transcriptional upregulation was massive (~600 fold). Can the authors explain this discrepancy? Could it be an antibody issue? It's also misleading, somewhat, to represent the data here as absolute numbers where every other figure has quantified similar data as a percentage.

Reviewer #3:

Remarks to the Author:

The authors have done a very thorough and impressive revision, and have addressed all of the points I have raised. Importantly, my concerns regarding the novelty of their results are sufficiently addressed.

I have one comment regarding the comparison to previous work: I appreciate the time you took to compare the RNA profiles you have found to previously reported related RNA profiles, however, presenting the results in a table is very hard to interpret, thus an additional graph representation (e.g. heat map) will be more suitable.

Manuscript NCOMMS-18-08965A. Response to the reviewers

The authors would like to thank the editor and the reviewers for their positive feedback.

Based on the reviewers' comments, we have added the following to the revised manuscript:

1. Textual changes to the results section to address the point of reviewer #2 regarding the discrepancy between RNA and protein levels of Atf3.
2. Graphs to demonstrate the overlap between protein expression levels of dVenus and Atf3 (**Supplementary Fig. 9**).
3. Heat maps for better visualization of the comparison between our RNA-sequencing differential gene expression profiles to previously reported transcriptional changes in the hippocampus or activated neuronal ensembles following various stimuli and/or time-points after fear conditioning (**Supplementary Data 4**).

In the revised manuscript text, textual changes have been indicated in **green**.

Please find below a point-by-point reply (in black) to the reviewers' comments (in red):

Reviewer #2 (Remarks to the Author):

The authors have added substantial new data that has significantly improved the original manuscript. The essential controls to validate the authors' conclusions are now included and alleviate my initial concerns.

We appreciate the reviewer's comments and positive feedback.

While I still feel there isn't much novelty of establishing the CREB transcriptional in engram consolidation, I do take the authors' points that the sustained time window of 24 hours is a surprising result and the genes identified in the data set can be followed up on by the field. I just had a minor point to clarify on the ATF3 protein data set in supplemental figure 9. It's surprising that the protein expression is so low (~4x less expression than Venus) give that the transcriptional upregulation was massive (~600 fold). Can the authors explain this discrepancy? Could it be an antibody issue? It's also misleading, somewhat, to represent the data here as absolute numbers where every other figure has quantified similar data as a percentage.

We thank the reviewer for bringing up this point. Indeed, we attribute the discrepancy between Atf3 RNA and protein levels as most likely due to the quality of the best available antibody (C-19, sc-188, SantaCruz) for immunohistochemical labeling. Other non-mutually exclusive possibilities include the absolute abundance of Atf3 RNA and the regulation of Atf3 RNA translation. Notably however, and consistent with the strong enrichment of Atf3 within the DG memory engram, across all experimental conditions, nearly every Arc+ / dVenus+ cell exhibited co-expression of Atf3. We have now modified the results section of the article to reflect these possibilities and included quantifications of dVenus and Atf3 expression overlap, as suggested by the reviewer, to Supplementary Figure 9.

Reviewer #3 (Remarks to the Author):

The authors have done a very thorough and impressive revision, and have addressed all of the points I have raised. Importantly, my concerns regarding the novelty of their results are sufficiently addressed.

We greatly appreciate the reviewer's acknowledgement of our contribution.

I have one comment regarding the comparison to previous work: I appreciate the time you took to compare the RNA profiles you have found to previously reported related RNA profiles,

however, presenting the results in a table is very hard to interpret, thus an additional graph representation (e.g heat map) will be more suitable.

We have now added heatmaps to Supplementary Data 4 in order to facilitate visualization of the comparisons between our current results and those of earlier studies.